# ROBUST IMITATION VIA MIRROR DESCENT INVERSE REINFORCEMENT LEARNING

## ABSTRACT

Adversarial imitation learning techniques are based on modeling statistical divergences using agent and expert demonstration data. However, unbiased minimization of these divergences is not usually guaranteed due to the geometry of the underlying space. Furthermore, when the size of demonstrations is not sufficient, estimated reward functions from the discriminative signals become uncertain and fail to give informative feedback. Instead of formulating a global cost at once, we consider reward functions as an iterative sequence in a proximal method. In this paper, we show that rewards dervied by mirror descent ensures minimization of a Bregman divergence in terms of a rigorous regret bound of $\mathcal{O}(1/T)$ for a particular condition of step sizes $\{\eta_t\}_{t=1}^T$. The resulting mirror descent adversarial inverse reinforcement learning (MD-AIRL) algorithm gradually advances a parameterized reward function in an associated reward space, and the sequence of such functions provides optimization targets for the policy space. We empirically validate our method in discrete and continuous benchmarks and show that MD-AIRL outperforms previous methods in various settings.

## 1 INTRODUCTION

Inverse reinforcement learning (IRL) is an algorithm of learning ground-truth rewards from demonstrations of an expert that acts optimally with respect to an unknown reward function. Traditional IRL approaches (Ng & Russell, 2000; Abbeel & Ng, 2004; Ziebart et al., 2008) solve the imitation problem based on iterative algorithms, alternating between the reward estimation process and a reinforcement learning (RL) algorithm. Recent adversarial imitation learning (AIL) studies (Ho & Ermon, 2016; Fu et al., 2017) focus on formulating a statistical divergence minimization with a fine-tuned representation of the target expert probability distribution (Ghasemipour et al., 2020).

One method to approximate the distance between probability distributions is using Bregman divergences, a family of metric-like functions induced by a strongly convex function. Jeon et al. (2020) generalized the AIL framework to an optimization problem with respect to a Bregman divergence between policy distributions where the method shares on the essence of regularized Markov decision processes (Geist et al., 2019). Through the lens of differential geometries, this implies that a policy distribution and a reward function can be associated with geometric constraints specified by a convex regularizer (Shima, 2007). Apparently, the limitation of regularized IRL studies naturally comes from a geometric property in which minimizing a Bregman divergence does not guarantee unbiased progression due to the constraints of the underlying space (Butnariu & Resmerita, 2006).

The success of machine learning has been the key to learning such divergences with high precision. When the demonstration size is sufficiently large, the discriminator approximation of AIL can model the entire expert policy distribution internally using nonlinear approximators such as neural networks (Finn et al., 2016; Fu et al., 2017). However, there are challenging tasks that require long action sequences to be solved. When states can only be rarely visited by imitation learning agents, the uncertainty of estimated expert policy often substantially affects the discrimination quality. Therefore, the development of a robust IRL method that is tolerant to unreliable discriminative signals induced by imperfect demonstrations is necessary.

The reasonable question is how to learn with cost estimations that are not precisely determined throughout the learning process. There have been considerable achievements in dealing with temporal costs in the *online learning* domain (Fiat & Woeginger, 1998; Hazan, 2019). An online learning

algorithm predicts a sequence of parameters $\{w_t\}_{t=1}^T \subset \mathcal{W}$ that is incurred by a cost function $F_t(\cdot)$ at each iteration. The most ordinary approach in online learning is stochastic gradient descent (SGD): $w_{t+1} = w_t - \eta_t \nabla F_t(w_t)$. SGD is a desirable algorithm when the set $\mathcal{W}$ is the Euclidean geometry since it can provide unbiased progression for minimizing $F_t$ (Raskutti & Mukherjee, 2015). However, policies appear in manifolds of probability distributions; a gradient may not be the direction of the steepest descent in this case due to geometric constraints (Amari, 1998; 2016).

In this paper, we propose an iterative imitation learning framework that interprets the RL-IRL scheme similar to proximal optimization methods (Boyd et al., 2004; Amari, 2016). We identify two issues in the AIL studies that have characteristics of unconstrained updates: (1) a divergence does not guarantee informative global directions to match the expert policy due to the geometry, and (2) a representation of divergence often cannot be accurately obtained due to insufficient data.

Our method is motivated by a template of optimization algorithms specified by a convex function, called mirror descent (MD; Nemirovsky & Yudin 1983). For parameters sequences, cost functions, and step sizes, an MD update for a strongly convex function $\Omega$ is derived as

$$\nabla\Omega(w_{t+1}) = \nabla\Omega(w_t) - \eta_t \nabla F_t(w_t), \tag{1}$$

where $\nabla\Omega$ is a bijective transformation that links the primal space and the dual space of gradients. Different from these standard MD formulations, our methodology draws a sequence of functions on a space formulated by a regularized reward operator which is an alternative to the dual space. Consequently, the reward functions are projected optimization targets for the space of policies.

**Our Contributions.** We propose a novel IRL algorithm which facilitates the agent in robustly imitating the expert. Our work is complementary to previous regularized IRL studies; we introduce a geometric perspective for optimizing rewards and derive solutions with theoretical guarantees.

NEW **R1**, **R2**

- Instead of a monolithic estimation process of a global solution, we derive a sequence of reward functions that provides local optimization targets for the space of policies (Section 3).
- We prove that rewards derived by an MD algorithm guarantee convergent divergence minimization performance along with a rigorous regret bound (Section 4).
- We propose mirror descent adversarial inverse reinforcement learning (MD-AIRL), a novel IRL algorithm that can be easily implemented on top of the standard AIL framework (Section 5).
- We validate the outperforming performance of MD-AIRL on benchmarks with large discrete action spaces and continuous action spaces (Section 6).

## 2 PRELIMINARIES

**Notation.** For finite sets $X$ and $Y$, we define $Y^X$ as a set of functions from $X$ to $Y$. $\Delta_X$ $(\Delta_X^Y)$ is a set of (conditional) probabilities over $X$ (conditioned on $Y$). For a function $f \in R^{\mathcal{S}\times\mathcal{A}}$ and a policy $\pi \in \Delta_{\mathcal{A}}^{\mathcal{S}}$, $f^s$ and $\pi^s$ denote shorthand notation of $f(s,\cdot)$ and $\pi(\cdot|s)$, respectively. A Markov decision process (MDP) is defined as a tuple $(\mathcal{S}, \mathcal{A}, P, r, \gamma)$ with the state space $\mathcal{S}$, the action space $\mathcal{A}$, the transition kernel $P \in \Delta_{\mathcal{S}}^{\mathcal{S}\times\mathcal{A}}$, the reward function $r \in \mathbb{R}^{S\times A}$ and the discount factor $\gamma \in [0,1)$. A Bregman divergence with respect to a convex function $\Omega$ for a state $s$ is defined as $D_\Omega(\pi^s\|\hat\pi^s) = \Omega(\pi^s) - \Omega(\hat\pi^s) - \langle\nabla\Omega(\hat\pi^s), \pi^s - \hat\pi^s\rangle_{\mathcal{A}}$. We write $\Omega(\pi) = [\Omega(\pi^s)]_{s\in\mathcal{S}}$ and similarly for $\nabla\Omega$.

**Regularized RL & IRL.** We consider the RL-IRL framework in regularized MDPs (Geist et al., 2019), where the policy is optimized along with a causal convex regularizer. The objective is to find $\pi$ which maximizes the expected discounted sum of rewards with a strongly convex regularizer $\Omega$:

$$\underset{\pi\in\Delta_{\mathcal{A}}^{\mathcal{S}}}{\text{maximize}} \, J_\Omega(\pi, r) := \mathbb{E}_\pi\left[\sum_{i=0}^\infty \gamma^i\{r(s_i, a_i) - \Omega(\pi(\cdot|s_i))\}\right], \tag{2}$$

where the subscript $\pi$ on the expectation denotes samples generated from the MDP and policy $\pi$.

Consider the convex conjugate of $q_E^s = \nabla\Omega(\pi_E^s)$: $\Omega^*(q_E^s) = \max_{\pi^s\in\Delta_{\mathcal{A}}}\langle\pi^s, q_E^s\rangle_{\mathcal{A}} - \Omega(\pi^s)$. Differentiating both sides with respect to $q_E^s$, the gradient of conjugate $\nabla\Omega^*$ maps $q_E$ to the unique optimal policy $\pi_E$. Let $\Pi$ be a bounded, open set for $\Delta_{\mathcal{A}}^{\mathcal{S}} \subset \Pi$ and $R : \Pi \to \mathbb{R}^{\mathcal{S}\times\mathcal{A}}$ be a reward operator which maps a policy to a reward function. The output $\hat r = R(\pi_E)$ is a representation of the expert's behavior; but finding such operator is an ill-posed problem because every function $\hat r$ that makes the state-action value function $q_E$ is a valid solution of regularized IRL (Geist et al., 2019). Recently, Jeon et al. (2020) proposed a tractable solution, which we refer to the function as the regularized reward function. We rewrite the previous notation by defining the *regularized reward operator* $\Psi_\Omega$.

**Definition 1** (Regularized reward operators). *Define the regularized reward operator $\Psi_\Omega : \pi \mapsto \psi_\pi$*

$$\psi_\pi(s,a) \coloneqq \Omega'(s,a;\pi) - \langle \pi^s, \nabla\Omega(\pi^s) \rangle_\mathcal{A} + \Omega(\pi^s) \qquad \forall s \in \mathcal{S}, a \in \mathcal{A}, \qquad (3)$$

*for* $\Omega'(s,\cdot;\pi) \coloneqq \nabla\Omega(\pi^s) = [\nabla_p \Omega(p)]_{p=\pi(\cdot|s)}$.

By using the operator in discriminative networks, Jeon et al. (2020) proposed regularized adversarial IRL (RAIRL) as a generalization of AIL in terms of minimizing a specific Bregman divergence.

**Mirror Descent.** Let $w$ be a parameter on a set $\mathcal{W}$ and $F_t : \mathcal{W} \to \mathbb{R}$ be a convex cost function from a class of functions $\mathcal{F}$. Replacing the L2 proximity term of proximal gradient descent with the Bregman divergence (Gutman & Peña, 2018), the proximal form of the MD update is given as

$$w_{t+1}^{\mathrm{MD}} = \underset{w \in \mathcal{W}}{\mathrm{argmin}} \langle \nabla F_t(w_t^{\mathrm{MD}}), w - w_t^{\mathrm{MD}} \rangle_\mathcal{W} + \alpha_t D_\Omega(w \| w_t^{\mathrm{MD}}), \qquad (4)$$

where $\alpha_t \coloneqq 1/\eta_t$ denotes an inverse of the step size $\eta_t$. MD was developed as a generalization of SGD, where the local geometry is specified by a Bregman divergence (Gunasekar et al., 2020).

## 3 ITERATIVE RL-IRL AS A PROXIMAL OPTIMIZATION METHOD

We interpret the RL-IRL framework as a variant of proximal optimization methods and consider two sequences $\{\pi_t\}_{t=1}^\infty$ and $\{\psi_t\}_{t=1}^\infty$ that denote the learning policies and reward functions, respectively.

**Associated reward function.** An updated point in MD can be uniquely projected to the desired space ($\Delta_\mathcal{A}^\mathcal{S}$ in our case) using a Bregman projection operator $\mathcal{P}_\Omega$ that locates iterative points to the feasible region, i.e. $\pi_{t+1} = \mathcal{P}_\Omega(\tilde{\pi}_{t+1}) \coloneqq \mathrm{argmin}_{\pi \in \Delta_\mathcal{A}^\mathcal{S}} [D_\Omega(\pi \| \tilde{\pi}_{t+1})]_{s \in \mathcal{S}}$ for $\tilde{\pi}_{t+1} \in \Pi$. To avoid these computations, we preemptively constrain a reward operator to satisfy the projection invariance with respect to regularized RL process of the associated state-action value function $\hat{q}$: $\mathcal{P}_\Omega(\nabla\Omega^*(\hat{q})) = \nabla\Omega^*(\hat{q})$. According to Lemma 1 of Jeon et al. (2020), a regularized reward function $\psi_\pi$ can replace $\hat{q}$, since the induced Bregman divergence allows the learning in a greedy manner. **FIX R3** The projection invariance of the operator $\Psi_\Omega$ can be shown by a bijective relation with $\nabla\Omega^*$ in the space of $\Delta_\mathcal{A}^\mathcal{S}$ by the following lemma.

**Lemma 1** (Natural isomorphism). *Let $\psi \in \Psi_\Omega(\Delta_\mathcal{A}^\mathcal{S})$ for $\Psi_\Omega(\Pi) = \{ \psi \mid \psi(s,a) = \psi_\pi(s,a), \forall s \in \mathcal{S}, a \in \mathcal{A}, \pi \in \Pi \}$. Then, $\nabla\Omega^*(\psi)$ is unique with respect to $\psi$, and for every $\pi = \nabla\Omega^*(\psi)$, $\pi \in \Delta_\mathcal{A}^\mathcal{S}$.*

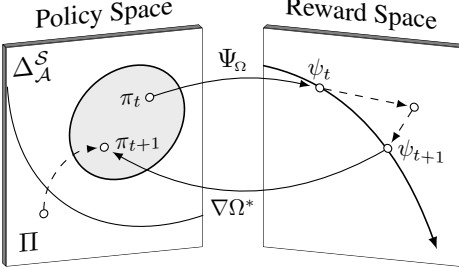

**FIX R3** Figure 1: A schematic illustration. Suppose an update is constrained by a divergence with respect to current $\pi_t$ (gray). MD is performed in the space defined by $\Psi_\Omega$. $\pi_{t+1}$ is recovered in the desired space $\Delta_\mathcal{A}^\mathcal{S}$ (solid curve) using $\nabla\Omega^*$.

The proof is in Appendix A. Figure 1 illustrates that there is unique $\psi_t$ for $\pi_t$, hence the two functions are isomorphic objects. Note that the transformed $\Psi_\Omega(\pi_t)$ is different from the form $\nabla\Omega(\pi_t)$ of the dual space that is used in classical MD algorithms. The transformation is shifted by a vector $\mathbf{1}c$ with a constant $c = \Omega(\pi_t^s) - \langle \pi_t^s, \nabla\Omega(\pi_t^s) \rangle_\mathcal{A}$ for each $s$. However, if the underlying space is $\Delta_\mathcal{A}^\mathcal{S}$, the operator $\nabla\Omega^*$ reconstructs the original point for both $\Psi_\Omega$ and $\nabla\Omega$, since the distributivity of $\Omega$ (Geist et al., 2019): $\Omega^*(y + \mathbf{1}c) = \Omega^*(y) + c$, holds (so $\nabla\Omega^*(y + \mathbf{1}c) = \nabla\Omega^*(y)$). Consequently, we may omit the projection phase (the **NEW R3** dotted angle in Figure 1) when the reward function satisfies the constraint of $\Psi_\Omega(\Delta_\mathcal{A}^\mathcal{S})$.

As a result, we consider an updated reward function $\psi_{t+1} \in \Psi_\Omega(\Delta_\mathcal{A}^\mathcal{S})$ as a projected target of MD, **FIX R3** which can be associated by parameterization of $\Delta_\mathcal{A}^\mathcal{S}$. For instance, a bijective mapping of $\psi_{t+1}$ can **NEW R3** represent a softmax policy for a discrete space, or a Gaussian policy for a continuous space, using its parameters. The subsequent RL process at $t$-th step becomes finding the next iteration $\pi_{t+1}$ by **NEW R3**

$$\underset{\pi \in \Delta_\mathcal{A}^\mathcal{S}}{\mathrm{maximize}}\, J_\Omega(\pi, \psi_{t+1}) = \mathbb{E}_\pi \Big[ \sum_{i=0}^\infty \gamma^i D_\Omega(\pi^{s_i} \| \pi_{t+1}^{s_i}) \Big], \qquad \pi_{t+1} = \nabla\Omega^*(\psi_{t+1}). \qquad (5) \quad \textbf{FIX R3}$$

The equation shows that an arbitrary regularized RL algorithm with the regularizer $\Omega$ forms a cumulative discounted sum of Bregman divergences, thus the mapping to $\pi_{t+1}$ can be achieved.

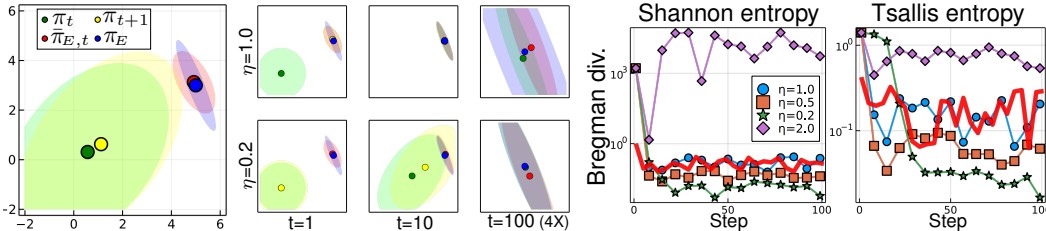

Figure 2: The policy $\pi_t$ minimizes an online cost $D_\Omega(\cdot\|\bar{\pi}_{E,t})$ associated with a convex regularizer such as negative Shannon and Tsallis entropies. The left example shows that the number of updates of $\pi_{t+1}$ vary by the step sizes $\eta$ of MD formulation. The plots on the right show that $D_\Omega(\pi_t\|\pi_E)$ vary by $\eta$ (averaged over 10 different trials). The solid red lines represent the baselines of $D_\Omega(\bar{\pi}_{E,t}\|\pi_E)$.

**Online imitation learning.** The necessity of online learning setup comes from observing that our RL-IRL processes do not retain $\psi_E = \Psi_\Omega(\pi_E)$, a representation of $\pi_E$, during training. Instead, consider a random process $\{\bar{\pi}_{E,t}\}_{t=1}^\infty$ where the estimation $\bar{\pi}_{E,t}$ resides in a closed, convex neighborhood of $\pi_E$, estimated by a separate estimation algorithm such as maximum likelihood methods. Substituting $\psi_E$ to $\psi_{\bar{\pi}_{E,t}}$ in $J_\Omega(\pi, \psi_E)$ turns the RL objective into an online learning problem:

$$\underset{\pi \in \Delta_\mathcal{A}^\mathcal{S}}{\text{minimize}}\, F_t(\pi) = -J_\Omega(\pi, \psi_{\bar{\pi}_{E,t}}) = \mathbb{E}_\pi\left[\sum\nolimits_{i=0}^\infty \gamma^i D_\Omega\left(\pi^{s_i}\big\|\bar{\pi}_{E,t}^{s_i}\right)\right]. \qquad (6) \quad \text{FIX} \;\; \textcolor{magenta}{\textbf{R3}}$$

For a deeper understanding of our setup, we consider an example in Figure 2. Assume that policies of the learning agent and the expert follow multivariate Gaussian distributions initially at $\pi_1 = \mathcal{N}([0,0]^\mathsf{T}, \mathbf{I})$ and $\pi_E = \mathcal{N}([5,3]^\mathsf{T}, \Sigma_E)$ for $|\Sigma_E| < 1$. The policy $\pi_t$ is trained by a cost function $D_\Omega(\cdot\|\bar{\pi}_{E,t})$ with the proximity constraint $\frac{1}{\eta}D_\Omega(\cdot\|\pi_t)$ where the Gaussian policy $\bar{\pi}_{E,t}$ is also fitted using stochastic samples from $\pi_E$ at each iteration, starting from $\bar{\pi}_{E,1} = \pi_1$. We first observe that the step size constant $\eta$ affects the training speed in the early phase. Notably, the performance of certain cases exceeds the baselines of $D_\Omega(\bar{\pi}_{E,t}\|\pi_E)$ by choosing the step size effectively low as $\eta < 1$. This suggests that there is a clear advantage of the online imitation learning setup for unreliable $\bar{\pi}_{E,t}$.

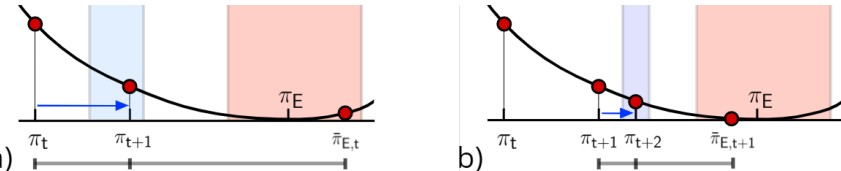

Figure 3: Illustrations of mirror descent imitation learning in the (a) $t$-th iteration and (b) $(t+1)$-th iteration. Consider that $\{\bar{\pi}_{E,t}\}_{t=1}^\infty$ is a random process and $\bar{\pi}_{E,t}$ and $\bar{\pi}_{E,t+1}$ are sampled from a neighborhood of $\pi_E$ with respect to a norm. The MD step is taken in the interval of $\pi_t$ and $\bar{\pi}_{E,t}$. Note that by decreasing the step size of updates, the region of $\pi_{t+1}$ (blue) shrinks.

**MD update rules.** Plugging each divergence of the cumulative cost $F_t$ to the template of Equation (4), the optimization process for the subsequent $\psi_{t+1}$ of the MD formulation is derived as[1]

$$\psi_{t+1} := \Psi_\Omega(\pi_{t+1}), \quad \pi_{t+1}^s = \underset{\pi^s \in \Delta_\mathcal{A}}{\arg\min} \big\langle \underbrace{\nabla D_\Omega(\pi_t^s\|\bar{\pi}_{E,t}^s)}_{\nabla\Omega(\pi_t^s) - \nabla\Omega(\bar{\pi}_{E,t}^s)}, \pi^s - \pi_t^s \big\rangle_\mathcal{A} + \alpha_t D_\Omega(\pi^s\|\pi_t^s)$$

$$= \underset{\pi^s \in \Delta_\mathcal{A}}{\arg\min}\, D_\Omega(\pi^s\|\bar{\pi}_{E,t}^s) - D_\Omega(\pi^s\|\pi_t^s) + \alpha_t D_\Omega(\pi^s\|\pi_t^s)$$

$$= \underset{\pi^s \in \Delta_\mathcal{A}}{\arg\min}\, \eta_t \underbrace{D_\Omega(\pi^s\|\bar{\pi}_{E,t}^s)}_{\text{estimated expert}} + (1-\eta_t)\underbrace{D_\Omega(\pi^s\|\pi_t^s)}_{\text{learning agent}}, \quad \eta_t = {1}/{\alpha_t}, \; \forall s \in \mathcal{S}, \quad (7)$$

where the gradient of $D_\Omega$ is taken with respect to its first argument $\pi_t^s$. The objective of reward learning is analogous to finding an interpolation at each iteration where the point is controlled by the step size $\eta_t$. Figure 3 shows that the region of $\pi_t$ (defined by a norm) gradually decreases

NEW $\;$ **R3**

---

[1]Bregman divergences are generally intractable to be computed, unless the policy is a specific parametric model (e.g. exponential families). See the works of Nielsen & Nock (2011); Jeon et al. (2020) and Appendix B.

when $\eta_t > \eta_{t+1}$. Note that solving the optimization of $F_t$ requires interaction between $\pi_t$ and the dynamics of the given environment; thus, the RL process in Equation (5) plays the essential role of sequential learning by the value measures of the reward function.

# 4 ANALYSES ON STEP SIZES, CONVERGENCE, AND REGRETS

**Online IRL.** We define an online cost $f_t(\pi_t, \tau_t)$ which involves a policy and a trajectory as inputs. The trajectory $\tau_t = \{s_i\}_{i=0}^{\infty}$ is available at the $t$-th step by executing the policy $\pi_t$ in the environment. An online cost function for reward $\bar{\psi}_{E,t} = \Psi_\Omega(\bar{\pi}_{E,t})$ can be expressed as

$$f_t(\pi_t, \tau_t) = \sum_{i=0}^{\infty} \gamma^i \Big\{ \Big( \sum_a \pi_t(a|s_i) \bar{\psi}_{E,t}(s_i, a) \Big) - \Omega\big(\pi_t(\cdot|s_i)\big) \Big\}. \tag{8} \quad \text{FIX} \quad \textbf{R3}$$

We refer the objective of the learning is to find a unique fixed point $\pi_* \in \Pi$ that minimizes $\mathbb{E}[f_t(\pi, \tau)]$, where the expectation is taken over the entire steps (i.e. $\lim_{t\to\infty} \mathbb{E}_{\tau_1,\cdots,\tau_t}[f(\pi_t, \tau_t)]$). $\quad$ NEW $\textbf{R3}$ Taking (stepwise) gradient for each $\pi(\cdot|s)$, $\pi_*$ is found by $\mathbb{E}[\nabla\Omega(\pi_*(\cdot|s)) - \nabla\Omega(\bar{\pi}_{E,t}(\cdot|s))] = 0$ when $t \to \infty$, hence $\nabla\Omega(\pi_*) = \lim_{t\to\infty} \mathbb{E}[\nabla\Omega(\bar{\pi}_{E,t}(\cdot|s))]$. The assumption of $\pi_E = \pi_*$ allows the particular situation when the estimation algorithm of $\bar{\pi}_{E,t}$ is actually convergent with $t \to \infty$. This also allows general situations where the estimated expert policy is not stationary; the algorithm finds the unique point according to the expectation of gradient by scheduling the step size $\eta_t$.

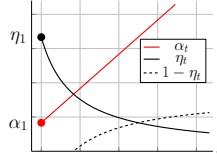

**Theoretical Analyses.** We state two conditions of $\{\eta_t\}_{t=1}^{\infty}$ to guarantee convergence properties which is explained in the following analyses.

- Convergent sequence & divergent series:

$$\lim_{t\to\infty} \eta_t = 0 \qquad \text{and} \qquad \sum_{t=1}^{\infty} \eta_t = \infty. \tag{9}$$

- Divergent series & convergent series of squared terms:

Figure 4: A step size sequence example.

$$\sum_{t=1}^{\infty} \eta_t = \infty \qquad \text{and} \qquad \sum_{t=1}^{\infty} \eta_t^2 < \infty. \tag{10}$$

Under the satisfaction of both cases, a sequence that is divergent in its series (e.g. harmonic series $\sum_{n=1}^{\infty} \frac{1}{n}$) is suitable as in Figure 4. In our arguments, a policy conditioned by a state is in a Banach space called the $L^p$ space ($\mathbb{R}^{\mathcal{A}}, \|\cdot\|_p$), where $\|\cdot\|_p$ denotes a $p$-norm, where we assume $1 < p \le 2$ for the convergence in the dual $L^q$ space ($1/p + 1/q = 1$). The proofs are in Appendix A.

**Theorem 1** (Stepsize considerations). *Let $\Omega$ be strongly convex, $\nabla\Omega$ be Lipschitz continuous, and the associated Bregman divergences are bounded. Assume $\inf_{\pi\in\Delta_{\mathcal{A}}^{\mathcal{S}}} \mathbb{E}[f_t(\pi, \tau)] > 0$. Then* $\lim_{T\to\infty} \mathbb{E}_{\tau_1,\cdots,\tau_T}\left[\sum_{i=0}^{\infty} D_\Omega(\pi_*^{s_i}\|\pi_T^{s_i})\right] = 0$ *if and only if Eq. (9) is satisfied.*

*(a) If $\lim_{t\to\infty} \eta_t = 0$, then $T \in \mathbb{N}$, $n < T$, and $c > 0$ exist s.t. $\mathbb{E}_{\tau_1,\cdots,\tau_T}\left[f_T(\pi_T, \tau_T)\right] \ge \frac{c}{T-n}$.*

*(b) If the step size takes the form $\eta_t = \frac{4}{t+1}$, then $\mathbb{E}_{\tau_1,\cdots,\tau_T}\left[\sum_{i=0}^{\infty} \gamma^i D_\Omega(\pi_*^{s_i}\|\pi_T^{s_i})\right] = \mathcal{O}(1/T)$.*

Theorem 2 addresses the convergence of the algorithm in a specific case when $\pi_E$ can be achieved.

**Theorem 2** (Convergence in optimal cases). *Let $\Omega$ be strongly convex, $\nabla\Omega$ be Lipschitz continuous, and the associated Bregman divergences are bounded. Assume $\pi_1 \ne \pi_E$ and $\inf_{\pi\in\Delta_{\mathcal{A}}^{\mathcal{S}}} \mathbb{E}[f_t(\pi, \tau)] = 0$. Then, $\lim_{t\to\infty} \mathbb{E}_{\tau_1,\cdots,\tau_t}\left[f_t(\pi_t, \tau_t)\right] = 0$ if and only if $\sum_{t=1}^{\infty} \eta_t = \infty$. If $\eta_t \equiv \eta_1$, then there exist $c_1, c_2 \in (0, 1)$ such that $c_1 \cdot A_1 \le A_T \le c_2 \cdot A_1$, where $\{A_t\}_{t=1}^{\infty}$ denote a sequence of $A_t = $ FIX $\textbf{R3}$ $\mathbb{E}_{\tau_1,\cdots,\tau_t}\left[D_\Omega(\pi_*\|\pi_t)\right]$.*

Proposition 1 provides a sufficient condition for the almost certain convergence of an MD algorithm by imposing the stronger condition of step size in Equation (10).

**Proposition 1** (Convergence in general cases). *Assume $\inf_{\pi\in\Delta_{\mathcal{A}}^{\mathcal{S}}} \mathbb{E}[f_t(\pi, \tau)] > 0$. If the step size sequence satisfies Eq. (10), then we have $\lim_{t\to\infty} \sum_{i=0}^{\infty} \gamma^i D_\Omega(\pi_*^{s_i}\|\pi_t^{s_i}) = 0$ almost surely.*

**Regrets.** Define the regret at $t$-th iteration for a sequence of cost functions $\{f_t\}_{t\in\mathbb{N}}$ as

$$\text{R}_t = \frac{1}{t}\sum_{j=1}^{t} f_j(\pi_j, \tau_j) - \inf_{\pi\in\Delta_{\mathcal{A}}^{\mathcal{S}}}\Big\{ \frac{1}{t}\sum_{j=1}^{t} f_j(\pi, \tau_j) \Big\}. \tag{11}$$

In the optimal case of $\inf_{\pi\in\Delta_{\mathcal{A}}^{\mathcal{S}}} \mathbb{E}[f_t(\pi, \tau_t)] = 0$, $\text{R}_t$ is bounded by $\mathcal{O}(1/T)$ since $f_t$ inherits the property of Bregman divergence so that the infimum is achieved by 0 at $\pi_E$. By Proposition 1, the updates regarding the policy converge when the step size sequence abides by Equation (10). Therefore, even for the general case of $\inf_{\pi\in\Delta_{\mathcal{A}}^{\mathcal{S}}} \mathbb{E}[f_t(\pi, \tau)] > 0$, $\text{R}_T$ is bounded to $\mathcal{O}(1/T)$.

---

**Algorithm 1** Mirror Descent Adversarial Inverse Reinforcement Learning.

---

1: **Input:** an expert trajectory dataset $(\tau_t^*)_{t=1}^T$, a regularized reward function $\psi_\phi \in \Psi_\Omega(\Delta_{\mathcal{A}}^{\mathcal{S}})$, an agent policy $\pi_\theta$, an estimate policy $\pi_\nu$, a neural network $b : \mathcal{S} \to \mathbb{R}$, $\lambda, \alpha_1, \alpha_T \in \mathbb{R}^+$.

2: **for** $t \leftarrow 1$ to $T$ **do**

3: $\quad \alpha_t \leftarrow \alpha_1 + \frac{\alpha_T - \alpha_1}{T-1}(t-1)$

4: $\quad$ Collect rollout trajectories $\tau_t$ by using the agent policy $\pi_\theta$.

5: $\quad$ Optimize $b$ via binary logistic regression for $D_b$ to classify $\tau_t^*$ and $\tau_t$.

6: $\quad$ Optimize $\pi_\nu$ via binary logistic regression for $D_{\theta,\nu}$ to classify $\tau_t^*$ and $\tau_t$.

7: $\quad$ Optimize $\psi_\phi$ with the MD objective in Equation (7) using $\alpha_t, \tau_t^*, \tau_t, \pi_\theta$, and $\pi_\nu$.

8: $\quad$ Train $\pi_\theta$ via a regularized actor-critic method to maximize $\psi_\phi^\lambda(s,a)$ with regularizer $\lambda\Omega(\cdot)$.

9: **Output:** $\pi_\theta, \psi_\phi^\lambda$.

---

# 5 MIRROR DESCENT ADVERSARIAL INVERSE REINFORCEMENT LEARNING

In this section, parameters $\theta$, $\phi$, and $\nu$ are presented representing agent policy, reward, and expert policy functions respectively parameterized with neural networks. On top of the standard AIL algorithm, we propose the MD-AIRL method, learning with a dual discriminator architecture motivated by previous studies regarding multiple discriminators (Nguyen et al., 2017; Chongxuan et al., 2017).

AIL methods have dealt divergences between joint densities of states and actions (Ghasemipour et al., 2020). To clearly distinguish matching overall state densities and imitating specific behavior, we aim to disentangle these concepts and propose two structured discriminators:

$$D_{\theta,\nu}(s,a) = \sigma\big(\log\{\pi_\nu(a|s)/\pi_\theta(a|s)\} + b(s)\big) \quad \text{and} \quad D_b(s) = \sigma(b(s)),$$

where $b : \mathcal{S} \to \mathbb{R}$ is a neural network regarding states and $\sigma(\cdot)$ denotes the sigmoid function. The discriminators $D_b$ and $D_{\theta,\nu}$ are trained with binary cross-entropy losses regarding trajectories:

$$\underset{\nu}{\text{maximize}}\, \mathbb{E}_{\pi_E}[\log D_{\theta,\nu}(s,a)] + \mathbb{E}_{\pi_t}[\log(1 - D_{\theta,\nu}(s,a))], \tag{12}$$

$$\underset{b}{\text{maximize}}\, \mathbb{E}_{\pi_E}[\log D_b(s)] + \mathbb{E}_{\pi_t}[\log(1 - D_b(s))], \tag{13}$$

Note that $b$ and $\theta$ are not trained for learning $D_{\theta,\nu}$. Let $\rho_\pi \in \Delta_{\mathcal{S}}$ denote normalized state visitation distribution of $\pi$, which is defined as $\rho_\pi(s) \coloneqq (1-\gamma)\mathbb{E}_\pi\big[\sum_{i=0}^\infty \gamma^i \mathbb{I}\{s_i = s\}\big]$ where $\mathbb{I}\{\cdot\}$ is an indicator function. Since $x \mapsto \alpha\log\sigma(x) + \beta\log(1-\sigma(x))$ attains its maximum at $\sigma(x) = \frac{\alpha}{\alpha+\beta}$ (Goodfellow et al., 2014), optimality is achieved when $\pi_\nu = \pi_E$ and $b(s) = \log \rho_{\pi_E}(s)/\rho_\pi(s)$.

Let $\psi_\phi$ denote the regularized reward function where the parameter $\phi$ is trained using a step size $\eta_t$, the agent policy $\pi_\theta$, and the estimation of expert policy $\pi_\nu$:

$$\underset{\phi}{\text{minimize}}\, \mathbb{E}_{s \sim \bar{\tau}_t}\Big[\eta_t D_\Omega(\pi_\phi(\cdot|s)\|\pi_\nu(\cdot|s)) + (1-\eta_t)D_\Omega(\pi_\phi(\cdot|s)\|\pi_\theta(\cdot|s))\Big], \tag{14}$$

where $\pi_\phi$ denotes the transformed policy $\nabla\Omega(\psi_\phi)$ and $\bar{\tau}_t$ denotes mini-batches of states using the both agent and expert trajectories. The algorithm adjusts the term $\eta_t = 1/\alpha_t$ with a harmonic progression by linearly increasing $\alpha_t$ by the range of $[\alpha_1, \alpha_T]$, which is derived from our analyses.

For a hyperparameter $\lambda > 0$, the MD-AIRL reward function is defined by a linear combination:

$$\psi_\phi^\lambda(s,a) = \lambda\psi_\phi(s,a) + b(s).$$

By using arbitrary regularized RL which uses $\lambda\Omega(\cdot)$ as the regularization function, the reward learning of $\psi_\phi^\lambda$ regarding the agent $\pi_\theta$ is decomposed into the following two terms:

$$\mathbb{E}_{\pi_\theta}\Big[\psi_\phi^\lambda(s,a) - \lambda\Omega(\pi_\theta(\cdot|s))\Big] = \lambda\mathbb{E}_{\pi_\theta}\Big[\psi_\phi(s,a) - \Omega(\pi_\theta(\cdot|s))\Big] - D_{\text{KL}}(\rho_{\pi_\theta}\|\rho_{\pi_E})$$

$$= -\lambda\mathbb{E}_{\pi_\theta}\Big[D_\Omega(\pi_\theta(\cdot|s)\|\pi_\phi(\cdot|s))\Big] - D_{\text{KL}}(\rho_{\pi_\theta}\|\rho_{\pi_E}),$$

Minimizing the first term of $\mathbb{E}_{\pi_\theta}\big[D_\Omega(\pi_\theta^s\|\pi_\phi^s)\big]$ represents the online learning of MD formulation. However, $\psi_\phi$ that is trained on $\bar{\tau}_t$ cannot cover the entire reachable states since the state visitation is heavily misaligned in challenging sequential decision problems. Therefore, we propose the second auxiliary term of generalized KL divergence $D_{\text{KL}}(\rho_{\pi_\theta}\|\rho_{\pi_E})$, playing an additional role of facilitating the supports of state visitation densities to be matched properly. Algorithm 1 summarizes the entire procedure of MD-AIRL. See Appendix C for detailed implementation.

Table 1: Bregman divergences with ground-truth distribution with five different types of regularization. The numbers are multiplied by the dimension of action space (we report $|\mathcal{A}| \cdot D_\Omega(\pi_T \| \pi_E)$).

| Algorithm | $|\mathcal{A}| = 10^2$ | | $|\mathcal{A}| = 10^3$ | | $|\mathcal{A}| = 10^4$ | |
|---|---|---|---|---|---|---|
| | RAIRL | MD-AIRL | RAIRL | MD-AIRL | RAIRL | MD-AIRL |
| Shannon | $2.55 \pm 1.59$ | $\mathbf{2.28 \pm 1.20}$ | $140.3 \pm 87.5$ | $\mathbf{125.3 \pm 61.8}$ | $\mathbf{5752.9 \pm 2986}$ | $5943.1 \pm 2967$ |
| Tsallis | $0.21 \pm 0.13$ | $\mathbf{0.11 \pm 0.04}$ | $0.55 \pm 0.13$ | $\mathbf{0.24 \pm 0.03}$ | $4.95 \pm 2.3$ | $\mathbf{4.21 \pm 0.16}$ |
| exp | $0.27 \pm 0.17$ | $\mathbf{0.13 \pm 0.06}$ | $0.55 \pm 0.12$ | $\mathbf{0.23 \pm 0.03}$ | $5.06 \pm 2.44$ | $\mathbf{4.97 \pm 0.69}$ |
| cos | $0.05 \pm 0.04$ | $\mathbf{0.02 \pm 0.01}$ | $0.03 \pm 0.02$ | $\mathbf{0.01 \pm 0.01}$ | $0.21 \pm 0.62$ | $\mathbf{0.05 \pm 0.05}$ |
| sin | $0.34 \pm 0.25$ | $\mathbf{0.12 \pm 0.04}$ | $3.82 \pm 3.46$ | $\mathbf{1.07 \pm 0.75}$ | $8.12 \pm 3.82$ | $\mathbf{7.59 \pm 1.02}$ |

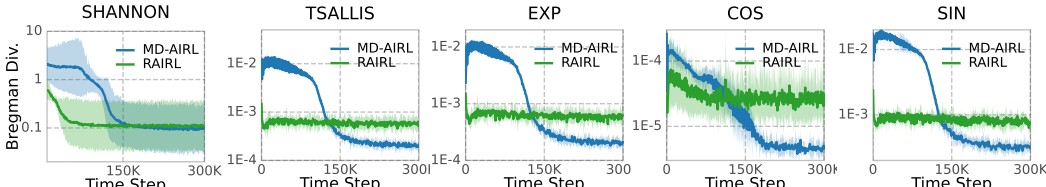

Figure 5: The average Bregman divergence measured on the log scale in multi-armed bandits at the action size of $|\mathcal{A}| = 10^3$. The shaded area represents 95% confidence interval for five runs.

## 6 EXPERIMENTAL RESULTS

For the RL algorithm, we implemented a RAC (Yang et al., 2019) method that is a generalization of SAC (Haarnoja et al., 2018) in terms of the regularization choice. We considered the class of separable regularizers $\Omega(p) = -\mathbb{E}_{a \sim p}[\varphi(p(a))]$: (1) Shannon entropy ($\varphi(x) = \log(x)$), (2) Tsallis entropy $\mathcal{T}_q$ ($\varphi(x; q) = \frac{1}{q-1}(x^{q-1} - 1)$, $q = 2$ by default), (3) exp regularizer ($\varphi(x) = e - e^x$), (4) cos regularizer ($\varphi(x) = \cos(\frac{\pi}{2}x)$), and (5) sin regularizer ($\varphi(x) = 1 - \sin\frac{\pi}{2}x$). We evaluated our approach on three topics (bandits, multiple goals, and MuJoCo environments). The main comparative method was RAIRL with a density-based model (RAIRL-DBM) since this model shares the identical level of expressiveness as our method when the parameterization of $\Delta_{\mathcal{A}}^{\mathcal{S}}$ is specified.

### 6.1 LARGE-SCALE MULTI-ARMED BANDITS

We first considered multi-armed bandit problems, where the cardinality of action spaces is varied. Learning the optimal distribution of $\pi_E$ becomes challenging as $|\mathcal{A}|$ increases, because the frequency of each sample becomes sparse due to the curse of dimensionality (Bellman et al., 1957). For each experiment, a stateless expert distribution $\pi_E$ was generated by the parameters of softmax distribution $\pi_E(i) = \exp(z_i)/\sum_j \exp(z_j)$ where the logits $z_i$ were randomly initialized to a uniform distribution. We set the action size to $|\mathcal{A}| = 10^2, 10^3, 10^4$ and restricted the sample size to 16.

Figure 5 shows that the Bregman divergence was large for MD-AIRL at the early training phase, because we chose the initial step size $\eta_1$ to be greater than 1 ($\alpha_1 = 0.5$). MD-AIRL exceeded the discriminative performance of RAIRL after certain steps, while the progression of RAIRL mostly stopped at local minima. Table 1 shows that MD-AIRL achieved overall lower Bregman divergence on average when three different cardinalities and five regularizers were considered. MD-AIRL outperformed RAIRL in four cases by choosing effectively low step size at the $\eta_T$ to be less than 1 ($\alpha_T = 2$). These results match properties of MD algorithms and our convergence analyses. Therefore, we argue that a constrained update rule with appropriate step sizes is necessary for robust reward acquisition and imitation for the situations when the total number of data samples is limited.

### 6.2 CONTINUOUS MULTI-GOAL ENVIRONMENT

We then considered an environment with a two-dimensional continuous state space. In this environment, an agent is a point mass initialized at the origin, and the four goals are located in the four cardinal directions. To draw meaningful reward surface, we considered multivariate Gaussian dis-

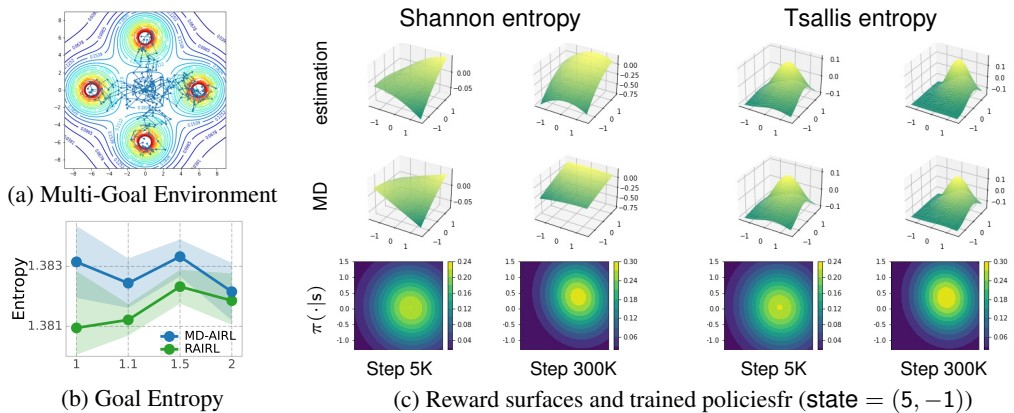

(a) Multi-Goal Environment

(b) Goal Entropy

(c) Reward surfaces and trained policiesfr (state $= (5, -1)$)

Figure 6: (a) The multi-goal environment, MD-AIRL trajectories, and the ground-truth rewards are shown. (b) The information entropies for the probabilities of achieving four goals. The x-axis indicates the $q$ value of the Tsallis entropy regularizers. The Shannon entropy regularizer is considered by the case of $q = 1$. (c) The top of each column shows regularized reward surfaces obtained by $\pi_\nu$. The middle and bottom show regularized rewards from $\pi_\phi$ and the policy $\pi_\theta$.

tribution policies with full covariance matrices. We parameterized a covariance matrix of Gaussian policy using lower triangular matrix which is an outcome of LDL decomposition (see Appendix C).

Figure 6 (a) shows trajectories generated by the trained agent. Figure 6 (b) shows that MD-AIRL achieved higher entropy for reaching the goals. Figure 6 (c) shows reward surfaces with regularizers, which was calculated by $\psi_\phi(s, a) + \varphi(\pi_\theta(a|s))$ for each point of $a \in \mathcal{A}$ and $s = (5, -1)$. During the training, the MD reward was similar to the estimated ground truth using adversarial training. However, the surface of MD-AIRL became flatter than the ground-truth estimation when $\pi_t$ was sufficiently close to the expert behavior. As a result, we claim that drastic changes in the target distribution, which are one of the typical characteristics of adversarial frameworks, are prevented. We argue that these characteristics mitigate overfitting caused by unreliable discriminatve signals.

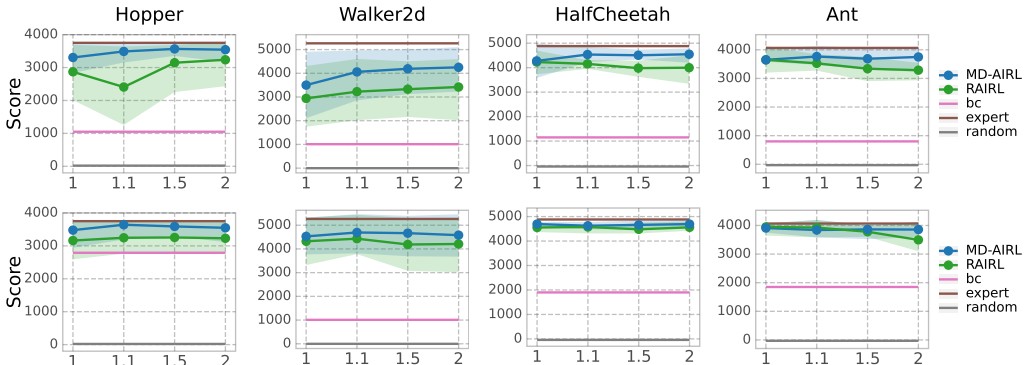

Figure 7: Average scores in MuJoCo benchmarks. The x-axis indicates the $q$ value of the Tsallis entropy regularizers. Shaded regions indicate 95% confidence intervals for four different runs. Top: 4 demonstrations. Bottom: 100 demonstrations.

## 6.3 CONTINUOUS CONTROL: MUJOCO

We validated MD-AIRL on MuJoCo continuous control tasks (Brockman et al., 2016). We assumed diagonal Gaussian policies for both learner's policy $\pi$ and expert policy $\pi_E$. Instead of the $\tanh$-squashed policy (Haarnoja et al., 2018), we used the hyperbolized environment assumption of RAIRL, which means that $\tanh$ is regarded as a part of the environment.

For each tasks, we considered two different numbers of episodes collected by an expert policy. In Figure 7, the performance of MD-AIRL, RAIRL, and behavior cloning (bc; Pomerleau, 1991) al-

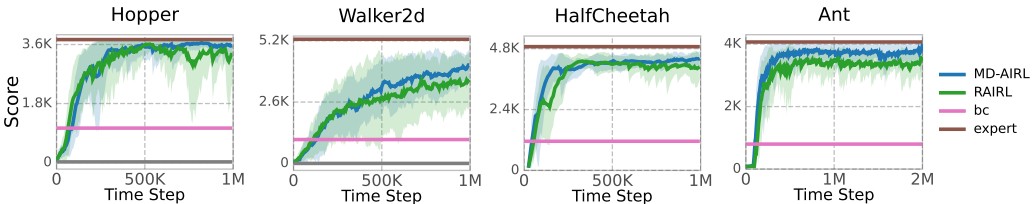

Figure 8: Average scores during training with 4 demonstrations (Tsallis regularizer $\mathcal{T}_q$ with $q = 2$).

gorithms are shown with the expert and random agent performance. In terms of sample efficiency, MD-AIRL outperformed RAIRL on `Hopper-v3`, `Walker-v3`, and `Ant-v3`. The performance gaps between MD-AIRL and RAIRL were more prominent in 4 episodes of expert demonstrations. The training curves in Figure 8 indicate that MD-AIRL showed lower variance than RAIRL especially after early phase of training. It can be concluded that MD-AIRL inherits the sample efficiency of AIL algorithms in challenging RL tasks benchmarks. Additionally, the algorithm is highly stable with respect to limited sample sizes, which is in alignment with our theoretical analyses.

## 7 RELATED WORKS

**Statistical manifolds.** The Hessian of strongly convex function forms a metric tenser of a manifold called Hessian geometries (Shima, 2007). Bregman divergences are similar to these metrics, providing useful metric-like properties (Butnariu & Resmerita, 2006). Using the Bregman divergence allows to solve many optimization problems, generalizing traditional approaches such as least squres (Boyd et al., 2004; Hiriart-Urruty & Lemaréchal, 2004). Probability distributions can be considered as points in a geometric space. A representative statistical manifold is the information geometry induced by the Fisher-Rao metric (Amari, 2016; Bauer et al., 2016; Nielsen, 2020).

**Regularized IRL.** Energy-based policies (i.e. Boltzmann distributions) have been appeared in early IRL researches (Ramachandran & Amir, 2007; Neu & Szepesvári, 2007; Babeş-Vroman et al., 2011). Notably, MaxEnt IRL (Ziebart et al., 2008; 2010) is a representative IRL algorithm based on information theory. Other statistical entropies have also been applied to the imitation learning problem, such as the Tsallis entropy derived from Tsallis statistics (Lee et al., 2018). Compared to RAIRL which also can use various convex regularizers (Jeon et al., 2020), our work allows more realistic situations where the expert policy cannot be precisely attained due to insufficient data, thanks to theoretical foundations originated from optimization studies.    NEW **R1**, **R2**

**Mirror descent.** MD is closely related to algorithms regarding non-Euclidean geometries with discretization of steps such as natural gradients (Amari, 1998; Raskutti & Mukherjee, 2015; Gunasekar et al., 2020). On the primal space, the infinitesimal limit of MD step corresponds to a Riemannian gradient flow (Do Carmo, 2016; Gunasekar et al., 2020). The online MD algorithms possess rigorous regret bounds (Srebro et al., 2011; Lei & Zhou, 2020); thus they can be highly efficient in terms of the number of evaluations until convergence.

## 8 DISCUSSION AND CONCLUSIONS

In this paper, we presented a novel IRL framework. We provided an mirror descent solution on reward functions and corresponding theoretical arguments. We proposed MD-AIRL, a practical adversarial IRL framework that can solve challenging imitation learning tasks. We verified that the proposed method has clear advantages over previous AIL methods in terms of robustness. As the reward hypothesis is grounded in obtaining a robust representation of the expert policy using IRL, we argued that current IRL studies lack robustness and theoretical guarantees for practical situations.    FIX **R3**

Considering RL and its inverse problem with geometric perspectives is vital for achieving desired goals in realistic situations. Although our work covers various online imitation learning methods with MD, it does not include some other cases when the proximity term is of other statistical divergence families such as f-divergence (Amari, 2016). Additionally, the boundedness and continuity assumptions on Bregman divergences in our analyses are usually justified, but outliers exist such as KL divergences. More sophisticated analyses on these general cases remain as future works.    NEW **R3**

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

# A  PROOFS

We define the space of policy as an object $\Delta_{\mathcal{A}}^{\mathcal{S}}$, which is a vector space formed by a collection of $|\mathcal{S}|$ elements of unit $(|\mathcal{A}| - 1)$-simplexes: $\Delta_{\mathcal{A}} = \left\{ x_1 e_1 + \cdots + x_{|\mathcal{A}|} e_{|\mathcal{A}|} \mid \sum_{i=1}^{|\mathcal{A}|} \theta_i = 1 \text{ and } \theta_i \geq 0 \text{ for } i \in \mathcal{A} \right\}$. We assume that each space of simplex is a subset of a specific Banach space called $L^p$ space $(\mathbb{R}^{\mathcal{A}}, \|\cdot\|)$, where $\|\cdot\|$ is a $p$-norm on $\mathcal{A}$. The dual space of $L^p$ space for $1 < p < \infty$ is $L^q$ space $(\mathbb{R}^{\mathcal{A}}, \|\cdot\|_*)$, where $\|\cdot\|_*$ is defined as a $q$-norm $(1/p + 1/q = 1)$. We assume $1 < p \leq 2$ for the convergence property in the dual $L^q$ space. We start with following definitions.

**Definition 2** (Lipschitz constants). *Given two metric spaces $(X, d_X)$ and $(Y, d_Y)$ where $d_X$ denotes the metric on set $X$ and $d_Y$ is the metric on set $Y$, a function $f : X \rightarrow Y$ is called Lipschitz continuous if there exists a real constant $k \geq 0$ such that, for all $x_1$ and $x_2$ in $X$,*

$$d_Y(f(x_1), f(x_2)) \leq k \cdot d_X(x_1, x_2). \tag{15}$$

*In particular, a function $f$ is called Lipschitz continuous if there exists a constant $k \geq 0$ such that,*

$$\|f(x_1) - f(x_2)\|_* \leq k \cdot \|x_1 - x_2\|, \quad \forall x_1, x_2 \tag{16}$$

*where norms $\|\cdot\|$ and $\|\cdot\|_*$ are endowed with each space $X$ and $Y$ respectively. For the smallest $L$ that substitutes $k$, $L$ is called the Lipschitz constant and $f$ is called the $L$-Lipschitz continuous function.*

**Definition 3** (Discrete-time martingale). *If a stochastic process $\{Z_t\}_{t \geq 1}$ satisfies $\mathbb{E}[|Z_n|] \leq \infty$ and*

*1)* $\mathbb{E}[Z_{n+1}|X_1, \ldots, X_n] \leq Z_n,$ *  2)* $\mathbb{E}[Z_{n+1}|X_1, \ldots, X_n] = Z_n,$ *  3)* $\mathbb{E}[Z_{n+1}|X_1, \ldots, X_n] \geq Z_n,$

*then, the stochastic process $\{Z_t\}_{t \geq 1}$ is called a 1) submartingale, 2) martingale, and 3) super-martingale, respectively, with respect to a filtration $\{X_t\}_{t \geq 1}$.*

The following analyses and proofs follow the results appeared in previous literatures for general aspects (Nemirovsky & Yudin, 1983; Gunasekar et al., 2020; Srebro et al., 2011; Lei & Zhou, 2020; Beck & Teboulle, 2003; Raskutti & Mukherjee, 2015). Our analyses extend existing results to imitation learning, and they are also highly general to cover various online methods for sequential decision problems.

## A.1  PROOF OF LEMMA 1

*Proof of Lemma 1.* The conjugate operator of $\psi_{\pi}^s$ satisfies the following identity (Lemma 1 of Jeon et al. (2020))

$$
\begin{aligned}
\Omega^*(\psi_\pi^s) &= \max_{\tilde{\pi}^s \in \mathbb{R}^{\mathcal{A}}} \langle \tilde{\pi}^s, \psi_\pi^s \rangle_{\mathcal{A}} - \Omega(\tilde{\pi}^s) \\
&= \max_{\tilde{\pi}^s \in \mathbb{R}^{\mathcal{A}}} \langle \tilde{\pi}^s, \nabla\Omega(\pi^s) \rangle_{\mathcal{A}} - \langle \pi^s, \nabla\Omega(\pi^s) \rangle_{\mathcal{A}} + \Omega(\pi^s) - \Omega(\tilde{\pi}^s) \\
&= \min_{\tilde{\pi}^s \in \mathbb{R}^{\mathcal{A}}} \Omega(\tilde{\pi}^s) - \Omega(\pi^s) - \langle \nabla\Omega(\pi^s), \tilde{\pi}^s - \pi^s \rangle_{\mathcal{A}} \\
&= \min_{\tilde{\pi}^s \in \mathbb{R}^{\mathcal{A}}} D_\Omega(\tilde{\pi}^s \| \pi^s)
\end{aligned}
$$

for every state $s \in \mathcal{S}$. By the property of Bregman divergence, and the convexity of $D_\Omega(\tilde{\pi}^s \| \pi^s)$ with respect to $\tilde{\pi}^s$ (Geist et al., 2019; Acharyya et al., 2013), the optimal condition is obtained by the unique maximizing argument $\tilde{\pi}(\cdot|s) = \pi(\cdot|s)$. By taking gradient to both sides with respect to $\psi_\pi^s$ we yield $\pi = \nabla\Omega^*(\psi_\pi)$.

If there is another $\tilde{\pi} \neq \pi$ that makes $\psi_{\tilde{\pi}} = \psi_\pi$, this contradicts the property of unique maximizing arguments for conjugates. Therefore, $\psi_\pi$ is uniquely defined for each $\pi$ and $\nabla\Omega^*(\psi) \in \Delta_{\mathcal{A}}^{\mathcal{S}}$. $\qquad\square$

We model the reward approximator regarding actions as $\psi \in \Psi_\Omega(\Delta_{\mathcal{A}}^{\mathcal{S}}) = \{\psi \mid \psi(s, a) = \psi_\pi(s, a), \forall s \in \mathcal{S}, a \in \mathcal{A}, \pi \in \Delta_{\mathcal{A}}^{\mathcal{S}}\}$, where $\psi_\pi$ is defined $\psi_\pi(s, a) := \Omega'(s, a; \pi) - \mathbb{E}_{a' \sim \pi(\cdot|s)}[\Omega'(s, a'; \pi)] + \Omega(\pi(\cdot|s))$. Note that $\Psi_\Omega(\pi)$ is different from $\nabla\Omega(\pi)$, but it is shifted by the amount of $-\mathbb{E}_{a' \sim \pi(\cdot|s)}[\Omega'(s, a'; \pi)] + \Omega(\pi)$ for every element of $\psi_\pi(s, \cdot)$. Therefore, the operators $\Psi_\Omega$ and $\nabla\Omega$ are link functions that form natural isomorphisms of functions.

## A.2 PROOF OF THEOREM 1

We consider the unique fixed point of $\pi_*$ as the solution of $\inf_{\pi \in \Delta_{\mathcal{A}}^{\mathcal{S}}} \lim_{t \to \infty} \mathbb{E}[f_t(\pi, \tau_t)]$ where the expectation indicates that we take consideration for all $\tau_t$ and $f_t$. By equating derivatives to zero, we write $\nabla\Omega(\pi_*) = \lim_{t \to \infty} \mathbb{E}[\nabla\Omega(\bar{\pi}_{E,t})])$. We provide more general results than previous imitation learning, as some of following statements include the case of $\inf_{\pi \in \Delta_{\mathcal{A}}^{\mathcal{S}}} \mathbb{E}[f_t(\pi, \tau)] > 0$, which means the estimates $\{\bar{\pi}_{E,t}\}_{t=1}^{\infty}$ do not converge to the fixed point of $\pi_*$, so $\lim_{t \to \infty} \|\pi_* - \bar{\pi}_{E,t}\| > 0$. Thus, MD-AIRL and other online MD algorithms allow realistic settings such as scarcity of data or imperfect demonstrations.

We first introduce a key relationship regarding cumulative gradients in our online MD setting.

**Lemma 2.** *Let $\{\pi_t\}_{t=1}^{\infty}$, $\{\bar{\pi}_{E,t}\}_{t=1}^{\infty}$, and $\{\eta_t\}_{t=1}^{\infty}$ be policy, estimate, and step size sequences, respectively. The subsequent policy $\pi_{t+1}$ in Equation (5) is obtained by an RL algorithm using the derivation of $\psi_{t+1}$ in Equation (7), resulting to the following equation:*

$$\pi_{t+1}(\cdot|s) = \operatorname*{argmin}_{\pi^s \in \Delta_{\mathcal{A}}} \eta_t D_\Omega(\pi^s \| \bar{\pi}_{E,t}^s) + (1 - \eta_t) D_\Omega(\pi^s \| \pi_t^s), \quad \forall s \in \mathcal{S}. \tag{17}$$

*We have for $t \in \mathbb{N}$,*

$$\eta_t \big(\nabla\Omega(\pi_t) - \nabla\Omega(\bar{\pi}_{E,t})\big) = \nabla\Omega(\pi_t) - \nabla\Omega(\pi_{t+1}) \tag{18}$$

*where we write $\nabla\Omega(\pi) = \big[\nabla\Omega(\pi(\cdot|s))\big]_{s \in \mathcal{S}}$*

*Proof of Lemma 2.* Since the optimization is convex with respect to each $\pi^s$, we equate the derivatives to 0 at $\pi_{t+1}$ as

$$\eta_t(\nabla\Omega(\pi_{t+1}^s) - \nabla\Omega(\bar{\pi}_{E,t}^s)) + (1 - \eta_t)\big(\nabla\Omega(\pi_{t+1}^s) - \nabla\Omega(\pi_t^s)\big) = 0, \quad \forall s \in \mathcal{S}.$$

Finally, we derive Equation (18) as

$$\eta_t(\nabla\Omega(\pi_{t+1}^s) - \nabla\Omega(\bar{\pi}_{E,t}^s)) + (1 - \eta_t)\big(\nabla\Omega(\pi_{t+1}^s) - \nabla\Omega(\pi_t^s)\big) = 0$$
$$\iff \nabla\Omega(\pi_{t+1}^s) - \eta_t \nabla\Omega(\bar{\pi}_{E,t}^s) - (1 - \eta_t)\nabla\Omega(\pi_t^s) = 0$$
$$\iff \nabla\Omega(\pi_t^s) - \nabla\Omega(\pi_{t+1}^s) = \eta_t\big(\nabla\Omega(\pi_t^s) - \nabla\Omega(\bar{\pi}_{E,t}^s)\big).$$

Therefore, the proof is complete. □

Lemma 2 indicates that the distances between dual maps are equivalent to $\eta_t \|\nabla\Omega(\bar{\pi}_{E,t}^s) - \nabla\Omega(\pi_t^s)\|_*$ Therefore, if the step size decreases as $\lim_{t \to \infty} \eta_t = 0$, $\lim_{t \to \infty} \|\nabla\Omega(\pi_t) - \nabla\Omega(\pi_{t+1})\|_* = 0$; thus, our argument in Section 3 is reasonable when $\Omega$ is strongly smooth.

Next, we reintroduce the three-point identity as follows.

**Lemma 3** (Three-point identity). *Let $\pi_a, \pi_b$, and $\pi_c$ be any policies with a given state. We have the following identity:*

$$\big\langle \nabla\Omega(\pi_a) - \nabla\Omega(\pi_b), \pi_c - \pi_b \big\rangle_{\mathcal{A}} = D_\Omega(\pi_c \| \pi_b) - D_\Omega(\pi_c \| \pi_a) + D_\Omega(\pi_b \| \pi_a).$$

*Proof of Lemma 3.* This can be derived using the definition of divergence as follows.

$$D_\Omega(\pi_c \| \pi_b) - D_\Omega(\pi_c \| \pi_a) + D_\Omega(\pi_b \| \pi_a) = \Omega(\pi_c) - \Omega(\pi_b) - \big\langle \nabla\Omega(\pi_b), \pi_c - \pi_b \big\rangle_{\mathcal{A}}$$
$$- \Omega(\pi_c) + \Omega(\pi_a) + \big\langle \nabla\Omega(\pi_a), \pi_c - \pi_a \big\rangle_{\mathcal{A}}$$
$$+ \Omega(\pi_b) - \Omega(\pi_a) - \big\langle \nabla\Omega(\pi_a), \pi_b - \pi_a \big\rangle_{\mathcal{A}}$$
$$= \big\langle \nabla\Omega(\pi_a) - \nabla\Omega(\pi_b), \pi_c - \pi_b \big\rangle_{\mathcal{A}}.$$

Therefore, the proof is complete. □

We now introduce basic identities regarding a Bregman divergence Lemmas 4 and 5 that are used to address progress of the algorithm.

**Lemma 4.** *Let $\pi_a, \pi_b$, and $\pi_c$ be any policies with a given state. The following identity holds.*

$$D_\Omega(\pi_c \| \pi_b) - D_\Omega(\pi_c \| \pi_a) = D_\Omega(\pi_a \| \pi_b) + \big\langle \nabla\Omega(\pi_a) - \nabla\Omega(\pi_b), \pi_c - \pi_a \big\rangle_{\mathcal{A}}. \tag{19}$$

*Proof of Lemma 4.* By Lemma 3, we have

$$D_\Omega(\pi_c\|\pi_b) - D_\Omega(\pi_c\|\pi_a) = -D_\Omega(\pi_b\|\pi_a) + \langle\nabla\Omega(\pi_a) - \nabla\Omega(\pi_b), \pi_c - \pi_b\rangle_\mathcal{A}.$$

Utilizing the identity of two Bregman divergences

$$D_\Omega(\pi, \tilde{\pi}) + D_\Omega(\tilde{\pi}, \pi) = \langle\nabla\Omega(\pi) - \nabla\Omega(\tilde{\pi}), \pi - \tilde{\pi}\rangle_\mathcal{A}, \tag{20}$$

we separate $\pi_c - \pi_b$ into $\pi_c - \pi_a$ and $\pi_a - \pi_b$ and write the rest of derivation as followings.

$$\begin{aligned} &D_\Omega(\pi_c\|\pi_b) - D_\Omega(\pi_c\|\pi_a) \\ &= \underbrace{-D_\Omega(\pi_b\|\pi_a) + \langle\nabla\Omega(\pi_a) - \nabla\Omega(\pi_b), \pi_a - \pi_b\rangle_\mathcal{A}}_{\text{Eq. (20)}} + \langle\nabla\Omega(\pi_a) - \nabla\Omega(\pi_b), \pi_c - \pi_a\rangle_\mathcal{A} \\ &= D_\Omega(\pi_a\|\pi_b) + \langle\nabla\Omega(\pi_a) - \nabla\Omega(\pi_b), \pi_c - \pi_a\rangle_\mathcal{A} \end{aligned}$$

Therefore, we obtain the desired identity. □

**Lemma 5.** *Let $\pi_a, \pi_b$, and $\pi_c$ be any policies with a given state. The following identity holds.*

$$D_\Omega(\pi_b\|\pi_a) - D_\Omega(\pi_c\|\pi_a) = -\langle\nabla\Omega(\pi_c) - \nabla\Omega(\pi_a), \pi_c - \pi_b\rangle_\mathcal{A} + D_\Omega(\pi_b\|\pi_c). \tag{21}$$

*Proof of Lemma 5.* By Lemma 3, we have

$$D_\Omega(\pi_b\|\pi_a) - D_\Omega(\pi_c\|\pi_a) = -D_\Omega(\pi_c\|\pi_b) + \langle\nabla\Omega(\pi_a) - \nabla\Omega(\pi_b), \pi_c - \pi_b\rangle_\mathcal{A}.$$

We separate $\nabla\Omega(\pi_a) - \nabla\Omega(\pi_b)$ into $\nabla\Omega(\pi_a) - \nabla\Omega(\pi_c)$ and $\nabla\Omega(\pi_c) - \nabla\Omega(\pi_b)$ and write the rest of derivation as followings.

$$\begin{aligned} &D_\Omega(\pi_b\|\pi_a) - D_\Omega(\pi_c\|\pi_a) \\ &= \underbrace{-D_\Omega(\pi_c\|\pi_b) + \langle\nabla\Omega(\pi_c) - \nabla\Omega(\pi_b), \pi_c - \pi_b\rangle_\mathcal{A}}_{\text{Eq. (20)}} + \langle\nabla\Omega(\pi_a) - \nabla\Omega(\pi_b), \pi_c - \pi_b\rangle_\mathcal{A} \\ &= D_\Omega(\pi_b\|\pi_c) + \langle\nabla\Omega(\pi_a) - \nabla\Omega(\pi_c), \pi_c - \pi_b\rangle_\mathcal{A} \end{aligned}$$

Therefore, we obtain the desired identity. □

We show the key arguments to prove Theorem 1 in the following lemma.

**Lemma 6.** *Assume $\inf_{\pi\in\Delta_\mathcal{A}^\mathcal{S}} \mathbb{E}[f_t(\pi, \tau)] > 0$. Assume also that $\Omega$ is $\omega$-strongly convex and $\nabla\Omega$ is L-Lipschitz continuous for $\omega \geq 0$ and $L \geq 0$. If $\lim_{t\to\infty} \mathbb{E}_{\tau_1,\cdots,\tau_t}\left[\sum_{i=0}^\infty \gamma^i D_\Omega\left(\pi_t^{s_i}\middle\|\pi_E^{s_i}\right)\right] = 0$ for $\pi_E$, then $\{\eta_t\}_{t=1}^\infty$ satisfies Equation (9). Furthermore, if $\Omega$ is strongly smooth, then Theorem 1 (a) holds with some constants $n \in \mathbb{N}$ and $c > 0$.*

*Proof of Lemma 6.* First, we show the condition of $\lim_{t\to\infty} \eta_t = 0$. Assuming all states are reachable (Fu et al. 2017, Definition B.1), the condition $\lim_{t\to\infty} \mathbb{E}_{\tau_1,\cdots,\tau_t}\left[\sum_{i=0}^\infty \gamma^i D_\Omega\left(\pi_t^{s_i}\middle\|\pi_E^{s_i}\right)\right] = 0$, **NEW R3**
implies $\lim_{t\to\infty} \mathbb{E}_{\tau_1,\cdots,\tau_t}\left[\|\pi_t - \pi_E\|\right] = 0$, where $\|\cdot\|$ is a matrix norm induced by the $p$-norm on $\mathcal{A}$. Then, our aim is to show that the gradient of strong convex function for $\pi_t$ converges to $\nabla\Omega(\pi_E)$ as

$$\lim_{t\to\infty} \mathbb{E}_{\tau_1,\cdots,\tau_t}\left[\left\|\nabla\Omega(\pi_t) - \nabla\Omega(\pi_E)\right\|_*\right] = 0. \tag{22}$$

To prove this point, we use the continuity of $\nabla\Omega$ at $\pi_E$; for any $\varepsilon > 0$, there exists some $0 < \delta \leq 1$ such that $\|\nabla\Omega(\pi) - \nabla\Omega(\pi_E)\|_* < \varepsilon$ whenever $\|\pi - \pi_E\| < \delta$.

When $\|\pi - \pi_E\| \geq \delta$, we apply the $L$-Lipschitz continuity assumption to find

$$\left\|\nabla\Omega(\pi) - \nabla\Omega(\pi_E)\right\|_* \leq L\|\pi - \pi_E\|, \tag{23}$$

where $\|\cdot\|_*$ is a matrix norm induced by the $q$-norm. Combining Equation (22) and Equation (23), we know that

$$\mathbb{E}_{\tau_1,\cdots,\tau_t}\left[\|\nabla\Omega(\pi_t) - \nabla\Omega(\pi_E)\|_*\right] \leq \varepsilon + L \cdot \mathbb{E}_{\tau_1,\cdots,\tau_t}\left[\|\pi_t - \pi_E\|\right]. \tag{24}$$

However, $\lim_{t\to\infty} \mathbb{E}_{\tau_1,\cdots,\tau_t}\left[\|\pi_E - \pi_t\|\right] = 0$ ensures the existence of some $n \in \mathbb{N}$ such that for $t > n$, there holds $\mathbb{E}_{\tau_1,\cdots,\tau_t}\left[\|\pi_E - \pi_t\|\right] < \varepsilon/L$. Applying this inequality to Equation (24), we have $\mathbb{E}_{\tau_1,\cdots,\tau_t}\left[\|\nabla\Omega(\pi_t) - \nabla\Omega(\pi_E)\|_*\right] < 2\varepsilon$ for some $t > n$.

For temporal estimation, let us denote the infimum of the expectation $\ell = \inf_{\pi \in \Delta_{\mathcal{A}}^{\mathcal{S}}} \mathbb{E}[\nabla\Omega(\pi_t) - \nabla\Omega(\bar{\pi}_{E,t})] > 0$. From Lemma 2, we have $\eta_t(\nabla\Omega(\pi_t) - \nabla\Omega(\bar{\pi}_{E,t})) = \nabla\Omega(\pi_t) - \nabla\Omega(\pi_{t+1})$. Taking the expectations on both sides of this equation yields

$$\eta_t \ell \leq \eta_t \mathbb{E}_{\tau_1,\cdots,\tau_{t+1}}\left[\|\nabla\Omega(\pi_{t+1}) - \nabla\Omega(\bar{\pi}_{E,t})\|_*\right] = \mathbb{E}_{\tau_1,\cdots,\tau_{t+1}}\left[\|\nabla\Omega(\pi_t) - \nabla\Omega(\pi_{t+1})\|_*\right].$$

Hence the convergence of the point $\nabla\Omega(\pi_t)$ is confirmed by taking the limit: $\lim_{t\to\infty} \eta_t = 0$.

Next, we show $\sum_{t=1}^{\infty} \eta_t = \infty$. By the $\omega$-strong convexity by the $L$-Lipschitz continuity of $\Omega$, we can find inequalities as

$$\left\langle \nabla\Omega(\pi) - \nabla\Omega(\tilde{\pi}), \pi - \tilde{\pi} \right\rangle_{\mathcal{A}} \leq L\|\pi - \tilde{\pi}\|^2 \leq \frac{2L}{\omega} D_\Omega(\pi\|\tilde{\pi}). \tag{25}$$

We note that $\|\pi_{t+1} - \bar{\pi}_{E,t}\| \leq \|\pi_t - \bar{\pi}_{E,t}\|$ so that there is a constant $\varepsilon$ that satisfies $\mathbb{E}[\|\pi_{t+1} - \bar{\pi}_{E,t+1}\|] \geq \mathbb{E}[\|\pi_{t+1} - \bar{\pi}_{E,t}\|] + \varepsilon$. Therefore, taking expectations in Equation (21) (setting $\pi_a = \bar{\pi}_{E,t}$, $\pi_b = \pi_{t+1}$, and $\pi_c = \pi_t$), for $\lim_{t\to\infty} \eta_t = 0$ and the strongly convex $\Omega$, we can find

$$\begin{aligned}
\mathbb{E}_{\tau_1,\cdots,\tau_{t+1}}\left[D_\Omega(\pi_{t+1}\|\bar{\pi}_{E,t+1})\right] &\geq \mathbb{E}_{\tau_1,\cdots,\tau_{t+1}}\left[D_\Omega(\pi_{t+1}\|\bar{\pi}_{E,t})\right] + \varepsilon' \\
&\geq (1 - a\eta_t)\mathbb{E}_{\tau_1,\cdots,\tau_t}\left[D_\Omega(\pi_t\|\bar{\pi}_{E,t})\right] + \mathbb{E}_{\tau_1,\cdots,\tau_{t+1}}\left[D_\Omega(\pi_{t+1}\|\pi_t)\right] + \varepsilon' \quad \text{(Eq. (21))} \\
&\geq (1 - a\eta_t)\mathbb{E}_{\tau_1,\cdots,\tau_t}\left[D_\Omega(\pi_t\|\bar{\pi}_{E,t})\right] + \varepsilon''
\end{aligned} \tag{26}$$

for some $t$ and $0 < \varepsilon' < \varepsilon''$. The positive constant $a = 2L/\omega$ is derived by the inequalities in Equation (25). We omit the conditioned states (superscripts) for simplicity of the derivation.

Since $\lim_{t\to\infty} \eta_t = 0$, we can also find a constant $n \in \mathbb{N}$ such that $\eta_t \leq (3a)^{-1}$ for $t \geq n$. Applying the inequality $1 - x > \exp(-2x)$ for $x \in (0, 1/3]$, we derive another inequality

$$\mathbb{E}_{\tau_1,\cdots,\tau_{t+1}}\left[D_\Omega(\pi_{t+1}\|\bar{\pi}_{E,t+1})\right] \geq \exp(-2a\eta_t)\mathbb{E}_{\tau_1,\cdots,\tau_t}\left[D_\Omega(\pi_t\|\bar{\pi}_{E,t})\right], \quad \forall t \geq n \tag{27}$$

Applying this for $t = T - 1, \ldots, n$ yields

$$\begin{aligned}
\mathbb{E}_{\tau_1,\cdots,\tau_T}\left[D_\Omega(\pi_T\|\bar{\pi}_{E,T})\right] &\geq \left[\prod_{t=n+1}^{T} \exp(-2a\eta_t)\right] \mathbb{E}_{\tau_1,\cdots,\tau_n}\left[D_\Omega(\pi_n\|\bar{\pi}_{E,n})\right] \\
&= \exp\left(-2a \cdot \sum_{t=n+1}^{T} \eta_t\right) \mathbb{E}_{\tau_1,\cdots,\tau_n}\left[D_\Omega(\pi_n\|\bar{\pi}_{E,n})\right].
\end{aligned} \tag{28}$$

Using Equation (28), we conclude $\mathbb{E}_{\tau_1,\cdots,\tau_n}\left[D_\Omega(\pi_{n+1}\|\bar{\pi}_{E,n+1})\right] > 0$. Otherwise, we have

$$\mathbb{E}_{\tau_1,\cdots,\tau_n}\left[D_\Omega(\pi_n\|\bar{\pi}_{E,n})\right] = \mathbb{E}_{\tau_1,\cdots,\tau_{n+1}}\left[D_\Omega(\pi_{n+1}\|\bar{\pi}_{E,n+1})\right] = 0$$

according to Equation (27), which leads to $\mathbb{E}_{\tau_1,\cdots,\tau_n}\left[\|\pi_n - \bar{\pi}_{E,n}\|^2\right] = \mathbb{E}_{\tau_1,\cdots,\tau_{n+1}}\left[\|\pi_{n+1} - \bar{\pi}_{E,n+1}\|^2\right] = 0$. This implies $\pi_n = \bar{\pi}_{E,n} = \pi_{n+1}$ almost surely, leading to $\mathbb{E}[f_t(\pi_t, \tau_t)] = 0$. This is a contradiction to the previous assumption $\inf_{\pi \in \Delta_{\mathcal{A}}^{\mathcal{S}}} \mathbb{E}[f_t(\pi, \tau_t)] > 0$, thus $\mathbb{E}_{\tau_1,\cdots,\tau_{n+1}}\left[D_\Omega(\pi_{n+1}\|\bar{\pi}_{E,n+1})\right] > 0$. Let us suppose the ideal case that the estimation process learns the exact $\pi_E$ in $t \to \infty$. In order to satisfy the limit $\lim_{T\to\infty} \mathbb{E}_{\tau_1,\cdots,\tau_T}\left[D_\Omega(\pi_T\|\bar{\pi}_{E,T})\right] = 0$ we see from Equation (28) that $\sum_{t=1}^{\infty} \eta_t = \infty$.

Now, we show that Theorem 1 (a) holds. Since $\Omega$ is $\omega$-strongly convex, so $\Omega^*$ is $(\omega^{-1})$-strongly smooth with respect to $\|\cdot\|_*$. Additionally, the $L$-Lipschitz continuity of $\nabla\Omega$ implies $L$-strong smoothness of $\Omega$; thus, naturally, $\Omega^*$ is $L$-strongly convex.

Combining these, for $\forall t \geq n$, the condition $\eta_t \leq (3a)^{-1}$ induces

$$\begin{aligned}
\mathbb{E}_{\tau_1,\cdots,\tau_{t+1}}\left[D_\Omega(\pi_{t+1}\|\bar{\pi}_{E,t+1})\right] \geq &(1 - a\eta_t)\mathbb{E}_{\tau_1,\cdots,\tau_t}\left[D(\pi_t\|\bar{\pi}_{E,t})\right] \\
&+ (2L)^{-1}\mathbb{E}_{\tau_1,\cdots,\tau_{t+1}}\left[\|\nabla\Omega(\pi_t) - \nabla\Omega(\pi_{t+1})\|_*^2\right],
\end{aligned}$$

and by Lemma 2, we get

$$\begin{aligned}
\mathbb{E}_{\tau_1,\cdots,\tau_{t+1}}\left[D_\Omega(\pi_{t+1}\|\bar{\pi}_{E,t+1})\right] \geq &(1 - a\eta_t)\mathbb{E}_{\tau_1,\cdots,\tau_t}\left[D_\Omega(\pi_t\|\bar{\pi}_{E,t})\right] \\
&+ (2L)^{-1}\eta_t^2 \mathbb{E}_{\tau_1,\cdots,\tau_{t+1}}\left[\|\nabla\Omega(\pi_t) - \nabla\Omega(\bar{\pi}_{E,t})\|_*^2\right].
\end{aligned}$$

Using the Cauchy-Schwarz inequality, we obtain a lower bound of the last term as

$$\mathbb{E}_{\tau_1,\cdots,\tau_t}\left[\|\nabla\Omega(\pi_t)-\nabla\Omega(\bar{\pi}_{E,t})\|_*^2\right] \geq \left\{\mathbb{E}_{\tau_1,\cdots,\tau_t}\left[\|\nabla\Omega(\pi_t)-\nabla\Omega(\bar{\pi}_{E,t})\|_*\right]\right\}^2 \geq \ell^2.$$

Thus, we obtain the final inequality as

$$\mathbb{E}_{\tau_1,\cdots,\tau_{t+1}}\left[D_\Omega(\pi_{t+1}\|\bar{\pi}_{E,t+1})\right] \geq (1-a\eta_t)\mathbb{E}_{\tau_1,\cdots,\tau_t}\left[D_\Omega(\pi_t\|\bar{\pi}_{E,t})\right] + (2L)^{-1}(\eta_t\ell)^2, \quad \forall t\geq n.$$

Applying this inequality from $t=T\geq n+1$ to $t=n+1$, we obtain

$$\mathbb{E}_{\tau_1,\cdots,\tau_{T+1}}\left[D_\Omega(\pi_{T+1}\|\bar{\pi}_{E,T+1})\right] \geq \mathbb{E}_{\tau_1,\cdots,\tau_n}\left[D_\Omega(\pi_n\|\bar{\pi}_{E,n})\right] \prod_{t=n+1}^{T}(1-a\eta_t)$$

$$+ (2L)^{-1}\ell^2 \sum_{t=n+1}^{T}\eta_t^2 \prod_{k=t+1}^{T}(1-a\eta_k)$$

$$\geq (2L)^{-1}\ell^2 \sum_{t=n+1}^{T}\eta_t^2 \prod_{k=t+1}^{T}(1-a\eta_k).$$

By the Cauchy-Schwarz inequality and our bound $0 < 1-a\eta_k \leq 1$ for $k\geq n$, we have

$$\sum_{t=n+1}^{T}\eta_t \prod_{k=t+1}^{T}(1-a\eta_k) \leq \left\{\sum_{t=n+1}^{T}\eta_t^2 \prod_{k=t+1}^{T}(1-a\eta_k)\right\}^{1/2}(T-n)^{1/2}.$$

Hence

$$\sum_{t=n+1}^{T}\eta_t^2 \prod_{k=t+1}^{T}(1-a\eta_k) \geq \frac{1}{a^2(T-n)}\left(\sum_{t=n+1}^{T}a\eta_t \prod_{k=t+1}^{T}(1-a\eta_k)\right)^2$$

$$= \frac{1}{a^2(T-n)}\left(\sum_{t=n+1}^{T}(1-(1-a\eta_t))\prod_{k=t+1}^{T}(1-a\eta_k)\right)^2$$

$$= \frac{1}{a^2(T-n)}\left(\sum_{t=n+1}^{T}\left[\prod_{k=t+1}^{T}(1-a\eta_k)-\prod_{k=t}^{T}(1-a\eta_k)\right]\right)^2$$

$$\geq \frac{1}{a^2(T-n)}\left(\sum_{t=n+1}^{T}1-\prod_{k=t}^{T}(1-a\eta_k)\right)^2$$

$$\geq \frac{1}{a^2(T-n)}(1-(1-a\eta_{n+1}))^2 = \frac{\eta_{n+1}^2}{T-n}.$$

Therefore, we obtain the lower bound of

$$\mathbb{E}_{\tau_1,\cdots,\tau_T}\left[D_\Omega(\pi_{T+1}\|\bar{\pi}_{E,T+1})\right] \geq \frac{\eta_{n+1}^2(2L)^{-1}\ell^2}{T-n}.$$

Since the Bregman divergence is bounded for all states, the sequence $\{\gamma^i D_\Omega(\pi_t^{s_i}\|\pi_t^{s_i})\}$ will converge as $i\to\infty$. Applying the monotone convergence theorem, we can interchange expectation and summation, which yields

$$\mathbb{E}_{\tau_1,\cdots,\tau_T}\left[\sum_{i=0}^{\infty}\gamma^i D_\Omega\big(\pi_T(\cdot|s_i)\|\bar{\pi}_{E,T}(\cdot|s_i)\big)\right] = \sum_{i=0}^{\infty}\mathbb{E}_{\tau_1,\cdots,\tau_T}\left[\gamma^i D_\Omega(\pi_T(\cdot|s_i)\|\bar{\pi}_{E,T}(\cdot|s_i))\right]$$

$$= \sum_{i=0}^{\infty}\gamma^i\mathbb{E}_{\tau_1,\cdots,\tau_T}\left[D_\Omega(\pi_T(\cdot|s_i)\|\bar{\pi}_{E,T}(\cdot|s_i))\right]$$

$$\geq \frac{\eta_{n+1}^2(2L-2L\gamma)^{-1}\ell^2}{T-n}, \quad \forall T\geq n.$$

This verifies Theorem 1 (a) with the constant $c = \eta_{n+1}^2(2L-2L\gamma)^{-1}\ell^2$. $\qquad\square$

Last, we show convergence to a unique fixed point of $\pi_*$ using the particular form of $\eta_t$ in Equation (9).

**Lemma 7.** *If $\{\eta_t\}_{t=1}^{\infty}$ satisfies Eq. (9), $\lim_{t\to\infty} \mathbb{E}_{\tau_1,\cdots,\tau_t}\left[\sum_{i=0}^{\infty} \gamma^i D_\Omega(\pi_*^s \| \pi_t^s)\right] = 0$. Furthermore, if the step size takes the form $\eta_t = \frac{4}{t+1}$, then $\mathbb{E}_{\tau_1,\cdots,\tau_T}\left[\sum_{i=0}^{\infty} \gamma^i D_\Omega\left(\pi_*^{s_i} \| \pi_T^{s_i}\right)\right] = \mathcal{O}(1/T)$.*

*Proof of Lemma 7.* According to Lemma 4 and the Bregman divergence of the conjugate $D_{\Omega^*}$, the one step progress regarding $\bar{\pi}_{E,t}$ can be written as

$$D_\Omega(\pi_* \| \pi_{t+1}) - D_\Omega(\pi_* \| \pi_t) = \langle \nabla \Omega(\pi_t) - \Omega(\pi_{t+1}), \pi_* - \pi_t \rangle_A + D_\Omega(\pi_t \| \pi_{t+1})$$
$$= \eta_t \langle \nabla \Omega(\pi_t) - \nabla \Omega(\bar{\pi}_{E,t}), \pi_* - \pi_t \rangle + D_{\Omega^*}(\nabla \Omega(\pi_{t+1}) \| \nabla \Omega(\pi_t)),$$

where we omit a given state. As $\omega$-strong convexity of $\Omega$ implies the $(\omega^{-1})$-strong smoothness of $\Omega^*$, we have

$$D_{\Omega^*}(\nabla \Omega(\pi_t) \| \nabla \Omega(\pi_{t+1})) \le \frac{1}{2\omega} \|\nabla \Omega(\pi_t) - \nabla \Omega(\pi_{t+1})\|_*^2 = \frac{\eta_t^2}{2\omega} \|\nabla \Omega(\pi_t) - \nabla \Omega(\bar{\pi}_{E,t})\|_*^2 \quad (29)$$

We bound $\|\nabla \Omega(\pi_t) - \nabla \Omega(\bar{\pi}_{E,t})\|_*^2$ by $2\|\nabla \Omega(\pi_t) - \nabla \Omega(\pi_*)\|_*^2 + 2\|\nabla \Omega(\pi_*) - \nabla \Omega(\bar{\pi}_{E,t})\|_*^2$, following the work of Lei & Zhou (2020). Since $\nabla \Omega$ is cocoercive with $\frac{1}{L}$ bye the Lipschitz continuity, we obtain

$$\|\nabla \Omega(\pi_t) - \nabla \Omega(\pi_*)\|_*^2 \le L \langle \nabla \Omega(\pi_*) - \nabla \Omega(\pi_t), \pi_* - \pi_t \rangle$$

thus

$$D_\Omega(\pi_* \| \pi_{t+1}) - D_\Omega(\pi_* \| \pi_t) \le \eta_t \langle \nabla \Omega(\pi_*) - \nabla \Omega(\bar{\pi}_{E,t}), \pi_* - \pi_t \rangle$$
$$- (1 - \frac{\eta_t L}{\omega})\eta_t \langle \nabla \Omega(\pi_*) - \nabla \Omega(\pi_t), \pi_* - \pi_t \rangle + \frac{\eta_t^2}{\omega}\left[\|\nabla \Omega(\pi_*) - \nabla \Omega(\bar{\pi}_{E,t})\|_*^2. \right. \quad (30)$$

By taking expectation, it follows that there exists $n \in \mathbb{N}$ such that $\eta_t \le \frac{\omega}{2L}$ for $t \ge n$ holds

$$\mathbb{E}_{\tau_t}\left[D_\Omega(\pi_* \| \pi_{t+1})\right] \le D_\Omega\left(\pi_* \| \pi_t\right) - \frac{\eta_t}{2} D_\Omega(\pi_* \| \pi_t) + z\eta_t^2, \quad (31)$$

where $z$ is the constant $z = \frac{1}{\omega}\mathbb{E}[\|\nabla \Omega(\pi_*) - \Omega(\bar{\pi}_{E,t})\|_*^2]$. Let $\{A_t\}_{t=1}^{\infty}$ denote a sequence of $A_t = \mathbb{E}_{\tau_1,\cdots,\tau_t}\left[D_\Omega(\pi_* \| \pi_t)\right]$. Then we have

$$A_{t+1} \le \left(1 - \frac{\eta_t}{2}\right) A_t + z\eta_t^2, \quad \forall t \ge n. \quad (32)$$

For a constant $h > 0$, we claim that $A_{t_1} < h$ for some $t_1 > n'$. Assume that this is not true, and we find some $t_2 \ge t_1$ such that $A_t > h, \forall t \ge t_2$. Since $\lim_{t\to\infty} \eta_t = 0$, there are some $t > t_3 > t_2$ that $\eta_t \le \frac{h}{4b}$. However, Equation (32) tells us that for $t \ge t_3$,

$$A_{t+1} \le \left(1 - \frac{\eta_t}{2}\right) A_t + z\eta_t^2 \le A_{t_3} - \frac{h}{4} \sum_{k=t'_\gamma}^{t} \eta_k \to -\infty \quad (\text{as } t \to \infty).$$

This is a contradiction, which verifies $A_t < h$ for $t > n'$. Since $\lim_{t\to\infty} \eta_t = 0$, we can find some $\eta_t$ that makes $A_t$ to be monotone decreasing. Then, we can conclude that the nonnegative sequence $\{A_t\}_{t=1}^{\infty}$ converges by iteratively applying the upper bounds.

We now prove Theorem 1 (b) under the condition and the choice $\eta_t = \frac{4}{t+1}$ of the step size sequence. The estimate becomes

$$A_{t+1} \le \left(1 - \frac{2}{t+1}\right) A_t + \frac{16z}{(t+1)^2}, \quad \forall t \ge n.$$

It follows the recurrence relation as

$$t(t+1)A_{t+1} \le (t-1)tA_t + 16z, \quad \forall t \ge n.$$

Applying this relation iteratively, we obtain the general form of inequality.

$$(T-1)TA_T \le (n-1)nA_n + 16z(T-n), \quad \forall T \ge n,$$

therefore we get the inequality as follows:

$$\mathbb{E}_{\tau_1,\cdots,\tau_T}\left[D_\Omega(\pi_* \| \pi_T)\right] \le \frac{(n-1)n\mathbb{E}_{\tau_1,\cdots,\tau_n}\left[D_\Omega(\pi_E \| \pi_n)\right]}{(T-1)T} + \frac{16z}{T}, \quad T \ge n.$$

By applying the monotone convergence theorem similar to Lemma 6, we conclude that $\mathbb{E}_{\tau_1,\cdots,\tau_T}[\sum_i^{\infty} \gamma^i D_\Omega(\pi_*(\cdot|s_i) \| \pi_T(\cdot|s_i))] = \mathcal{O}(1/T)$. $\square$

### A.3 PROOF OF THEOREM 2

**Necessity.** We rewrite the inequality in Equation (26) as

$$\mathbb{E}_{\tau_1,\cdots,\tau_{t+1}}\big[D_\Omega\big(\pi_{t+1}\big\|\bar\pi_{E,t+1}\big)\big] \geq (1 - 2L\omega^{-1}\eta_t)\mathbb{E}_{\tau_1,\cdots,\tau_t}\big[D_\Omega\big(\pi_t\big\|\bar\pi_{E,t}\big)\big]. \tag{33}$$

Since we assume that $\eta_t$ converge to 0 from previous arguments, onsider the step size sequence $0 < \eta_t \leq \frac{\omega}{(2+\kappa)L}$ for $\kappa > 0$ and $t \geq n$ where $n \in \mathbb{N}$. Denote a constant $\tilde a = \frac{2+\kappa}{2}\log\frac{2+\kappa}{\kappa}$ and apply the elementary inequality   FIX  **R3**

$$1 - x \geq \exp(-\tilde a x), \quad \forall 0 < x \leq \frac{2}{2+\kappa}$$

From Equation (33), it can be obtained that

$$\mathbb{E}_{\tau_1,\cdots,\tau_{t+1}}\big[D_\Omega\big(\pi_{t+1}\big\|\bar\pi_{E,t+1}\big)\big] \geq \exp\big(-2\tilde a L\omega^{-1}\eta_t\big)\mathbb{E}_{\tau_1,\cdots,\tau_t}\big[D_\Omega(\pi_t\|\bar\pi_{E,t})\big].$$

Applying this inequality iteratively for $t = n, \ldots, T-1$, then gives

$$\mathbb{E}_{\tau_n,\cdots,\tau_T}\big[D_\Omega(\pi_T\|\bar\pi_{E,T})\big] \geq \prod_{t=n}^{T-1}\exp\big(-2\tilde a L\omega^{-1}\eta_t\big)D_\Omega(\pi_n\|\bar\pi_{E,n})$$

$$= \exp\bigg\{-2\tilde a L\omega^{-1}\sum_{t=n}^{T-1}\eta_t\bigg\}D_\Omega(\pi_n\|\bar\pi_{E,n}).$$

From the assumption $\pi_E \neq \pi_n$, we have $D_\Omega(\pi_n\|\bar\pi_{E,n}) > 0$. The convergence $\lim_{t\to\infty}\mathbb{E}_{\tau_1,\cdots,\tau_t}\big[D_\Omega(\pi_t\|\bar\pi_{E,t})\big] = 0$ then implies $\sum_{t=1}^\infty \eta_t = \infty$.

**Sufficiency.** Here we use the estimate (32) derived in the proof of Lemma 7. However, in the optimal case, $z = \frac{1}{\omega}\mathbb{E}[f_t(\pi_*,\tau)] = 0$, so (32) takes the form (we can choose $n = 1$ by Equation (31)

$$A_{t+1} \leq \frac{\eta_t}{2}A_t, \quad \forall t \in \mathbb{N}. \tag{34}$$

This implies that for any $0 < h$, there must exist some integer $t_1 \in \mathbb{N}$ such that $A_t \leq \gamma$ for $t \geq t_1$. Otherwise, $A_t > \gamma$ for every $t \geq t_2$ with $t_2 \geq t_1$, which leads to a contradiction:

$$A_{t+1} \leq A_{t_2} - \frac{h}{2}\sum_{k=\tilde t_\gamma}^t \eta_k \to -\infty \text{ (as } t \to \infty).$$

Equation (34) also tells us that the sequence $\{A_t\}_{t=1}^\infty$ of nonnegative numbers is monotone decreasing. Hence $A_t \leq h$ for every $t \geq t_1$, which proves the convergence of $A_t$

$$\lim_{t\to\infty}\mathbb{E}_{\tau_1,\cdots,\tau_t}\big[D_\Omega(\pi_*\|\pi_t)\big] = \lim_{t\to\infty}A_t = 0.$$

We now prove the second point in Theorem 2 which is under the special choice of constant step size sequence $\eta_t \equiv \eta_1$. It follows from Equation (33) that $A_T \geq (1 - 2L\omega^{-1}\eta_1)^{T-1}A_1$. Hence, Eq (34) translates to

$$A_{t+1} \leq (1 - \eta_1/2)A_t,$$

from which we find $A_T \leq (1 - \eta_1/2)^{T-1}A_1$ by iteration starting from $t = 1$. This verifies the theorem with $c_1 = (1 - \frac{2L\eta_1}{\omega})$ and $c_2 = (1 - \frac{\eta_1}{2})$.

### A.4 PROOF OF PROPOSITION 1

The proof is based on Doob's forward convergence theorem.

**Theorem 3** (Doob's forward convergence theorem). *Let $\{X_t\}_{t\in\mathbb{N}}$ be a sequence of nonnegative random variables and let $\{\mathcal{F}_t\}_{t\in\mathbb{N}}$ be a filtration with $\mathcal{F}_t \subset \mathcal{F}_{t+1}$ for every $t \in \mathbb{N}$. Assume that $\mathbb{E}\big[X_{t+1}|\mathcal{F}_t\big] \leq X_t$ almost surely for every $t \in \mathbb{N}$. Then the sequence $\{X_t\}$ converges to a nonnegative random variable $X_\infty$ almost surely.*

We follow the proof of Lemma 7 and apply Eq. (30). Since $\langle \pi_* - \pi_t, \nabla\Omega(\pi_*) - \nabla\Omega(\pi_t)\rangle \geq 0$, Equation (30) implies

$$\mathbb{E}_{\tau_t}\left[D_\Omega(\pi_*\|\pi_{t+1})\right] \leq D_\Omega(\pi_*\|\pi_t) + \frac{\eta_t^2}{\omega}\mathbb{E}_{\tau_t}\left[\|\nabla\Omega(\pi_*) - \nabla\Omega(\bar{\pi}_{E,t})\|_*^2\right], \quad \forall t \geq n. \tag{35}$$

The condition $\sum_{t=1}^\infty \eta_t^2 < \infty$ enables us to define a stochastic process $\{X_t\}$ by

$$X_t = D_\Omega(\pi_*\|\pi_{t+1}) + \frac{1}{\omega}\mathbb{E}\left[\|\nabla\Omega(\pi_*) - \nabla\Omega(\bar{\pi}_{E,t})\|_*^2\right]\sum_{i=t+1}^\infty \eta_i^2.$$

By (35), we know that $\mathbb{E}_{\tau_t}[X_{t+1}] \leq X_t$ for $t \geq n$. Additionally, $X_t \geq 0$. Therefore, the stochastic process $\{X_t\}_{t-n+1\geq 1}$ is a submartingale (equivalently, $\{-X_t\}_{t-n+1\geq 1}$ is a supermartingale). Then by Lemma 3, the sequence $\{X_t\}_{t\geq 1}$ converges to a nonnegative random variable $X_\infty$ almost surely. According to Fatou's Lemma and the convergence $\lim_{t\to\infty}\mathbb{E}\left[\sum_{i=0}^\infty \gamma^i D_\Omega(\pi_*\|\pi_t)\right] = 0$ for any states a proved by Lemma 7, we obtain

NEW

$$\mathbb{E}[X] = \mathbb{E}\left[\lim_{t\to\infty}\sum_{i=0}^\infty \gamma^i D_\Omega\big(\pi_*(\cdot|s_i)\big\|\pi_t(\cdot|s_i)\big)\right] \leq (1-\gamma)^{-1}\liminf_{t\to\infty}\mathbb{E}\left[D_\Omega(\pi_*\|\pi_t)\right] = 0.$$

It follows that the sequence of discounted sum $\left\{\sum_{i=0}^\infty \gamma^i D_\Omega(\pi_*(\cdot|s_i)\|\pi_t(\cdot|s_i))\right\}_{t\in\mathbb{N}}$ converges to 0 almost surely.

## B  TSALLIS ENTROPY AND ASSOCIATED BREGMAN DIVERGENCE AMONG FULL COVARIANCE MULTIVARIATE GAUSSIAN DISTRIBUTIONS

In this section, we briefly reintroduce the tractable method to derive Bregman divergences and regularized reward functions proposed by Nielsen & Nock (2011) and Jeon et al. (2020). Then, we identify specific paremeterization to model Gaussian distributions with full covariance matrices.

The standard form of the exponential family is represented as

$$\exp\big\{\langle\theta, t(x)\rangle - F(\theta) + k(x)\big\}. \tag{36}$$

The parameterization is as follows:

$$\theta = \begin{bmatrix} \Sigma^{-1}\mu \\ -\frac{1}{2}\Sigma^{-1} \end{bmatrix} = \begin{bmatrix} \theta_1 \\ \theta_2 \end{bmatrix},$$

$$t(x) = \begin{bmatrix} x \\ xx^\mathsf{T} \end{bmatrix},$$

$$F(\theta) = -\frac{1}{4}\theta_1^\mathsf{T}\theta_2^{-1}\theta_1 + \frac{1}{2}\ln|-\pi\theta_2^{-1}| = \frac{1}{2}\mu^\mathsf{T}\Sigma^{-1}\mu + \frac{1}{2}\ln(2\pi)^d|\Sigma|,$$

$$k(x) = 0,$$

where we can analytically recover the multivariate Gaussian distribution (Nielsen & Nock, 2011)

$$\exp\big\{\langle\theta, t(x)\rangle - F(\theta) + k(x)\big\} \tag{37}$$

$$= \exp\left\{\mu^\mathsf{T}\Sigma^{-1}x - \frac{1}{2}\mathrm{tr}\big(\Sigma^{-1}xx^\mathsf{T}\big) - \frac{1}{2}\mu^\mathsf{T}\Sigma^{-1}\mu + \frac{1}{2}\ln(2\pi)^d|\Sigma|\right\} \tag{38}$$

$$= \frac{1}{(2\pi)^{d/2}|\Sigma|^{1/2}}\exp\left\{\mu^\mathsf{T}\Sigma^{-1}x - \frac{1}{2}x^\mathsf{T}\Sigma^{-1}x - \frac{1}{2}\mu^\mathsf{T}\Sigma^{-1}\mu\right\} \tag{39}$$

$$= \frac{1}{(2\pi)^{d/2}|\Sigma|^{1/2}}\exp\left\{\frac{1}{2}(x-\mu)^\mathsf{T}\Sigma^{-1}(x-\mu)\right\}. \tag{40}$$

For two distributions $\pi$ and $\hat{\pi}$ with $k(x) = 0$, Nielsen & Nock (2011) proposed the function $I(\cdot)$:

$$I(\pi, \hat{\pi}; \alpha, \beta) = \int \pi(x)^\alpha \hat{\pi}(x)^\beta \,\mathrm{d}x = \exp\left\{F\big(\alpha\theta + \beta\hat{\theta}\big) - \alpha F(\theta) - \beta F(\hat{\theta})\right\}$$

where the detailed derivation is as follows:

$$\int \pi(x)^\alpha \hat{\pi}(x)^\beta \, \mathrm{d}x$$

$$= \int \exp\Big\{\alpha\langle\theta, t(x)\rangle - \alpha F(\theta) + \beta\langle\hat{\theta}, t(x)\rangle - \beta F(\hat{\theta})\Big\} \, \mathrm{d}x$$

$$= \int \exp\Big\{\langle\alpha\theta + \beta\hat{\theta}, t(x)\rangle - F(\alpha\theta + \beta\hat{\theta})\Big\} \exp\Big\{F(\alpha\theta + \beta\hat{\theta}) - \alpha F(\theta) - \beta F(\hat{\theta})\Big\} \, \mathrm{d}x$$

$$= \exp\Big\{F(\alpha\theta + \beta\hat{\theta}) - \alpha F(\theta) - \beta F(\hat{\theta})\Big\} \int \exp\Big\{\langle\alpha\theta + \beta\hat{\theta}, t(x)\rangle - F(\alpha\theta + \beta\hat{\theta})\Big\} \, \mathrm{d}x$$

$$= \exp\Big\{F(\alpha\theta + \beta\hat{\theta}) - \alpha F(\theta) - \beta F(\hat{\theta})\Big\}.$$

### B.1   TSALLIS ENTROPY OF FULL COVARIANCE GAUSSIAN DISTRIBUTIONS

For $\varphi(x; q) = \frac{1}{q-1}(x^{q-1} - 1)$, the Tsallis entropy can be written as

$$\mathcal{T}_q(\pi) := -\mathbb{E}_{x\sim\pi}\varphi(x; q) = \int \pi(x)\frac{1 - \pi(x)^{q-1}}{q - 1} \, \mathrm{d}x$$

$$= \frac{1 - \int \pi(x)^q \, \mathrm{d}x}{q - 1} = \frac{1}{q - 1}\big(1 - I(\pi, \pi; q, 0)\big)$$

$$= \frac{1 - \exp\big(F(q\theta) - qF(\theta)\big)}{q - 1}.$$

If $\pi$ is a multivariate Gaussian distribution, we have

$$F(q\theta) = \frac{q}{2}\mu^\mathsf{T}\Sigma^{-1}\mu + \frac{1}{2}\ln(2\pi)^d|\Sigma| - \frac{1}{2}\ln q^d.$$

Since the covariance matrix is positive-definite, Cholesky decomposition can be applied, which separates the matrix to lower- and upper-triangle matrices. Likewise, we can apply LDL decomposition. Let us factorize the covariance $\Sigma = L\,\mathrm{diag}\{\sigma_1^2, \ldots, \sigma_d^2\}L^\mathsf{T}$ where $L$ denotes a unit lower triangular matrix produced by LDL decomposition. Then we have

$$F(q\theta) - qF(\theta) = (1 - q)\bigg\{\frac{d}{2}\ln 2\pi + \frac{1}{2}\ln|\Sigma| - \frac{d\ln q}{2(1 - q)}\bigg\}$$

$$= (1 - q)\bigg\{\frac{d}{2}\ln 2\pi + \frac{1}{2}\ln\prod_{i=1}^d \sigma_i^2 - \frac{d\ln q}{2(1 - q)}\bigg\}$$

$$= (1 - q)\sum_{i=1}^d \bigg\{\frac{\ln 2\pi}{2} + \ln\sigma_i - \frac{\ln q}{2(1 - q)}\bigg\}.$$

### B.2   TRACTABLE FORM OF $\psi_\pi$

For separable $\Omega$, $\psi_\pi$ is written as (Jeon et al., 2020)

$$\psi_\pi(s, a) = -f'(s, a) + \mathbb{E}_{a\sim\pi}[f'(\pi(a|s)) - \varphi(a|s)]$$

where $\varphi(x) = \frac{k}{q-1}(1 - x^{q-1})$ and accordingly $f(x) = x\varphi(x)$. For the gradient of $f(\cdot)$, we have

$$f'(x) = \frac{k}{q - 1}(1 - qx^{q-1})$$

$$= \frac{k}{q - 1}(q - qx^{q-1} - (q - 1))$$

$$= \frac{qk}{q - 1}(1 - x^{q-1}) - k$$

$$= q\varphi(x) - k.$$

Taking the expectation yields Tsallis entropy as follows.

$$\mathbb{E}_{x\sim\pi}\big[-f'(x;\pi)+\varphi(x)\big] = \mathbb{E}_{x\sim\pi}\big[k - q\varphi(x)+\varphi(x)\big] = (1-q)\mathcal{T}_q^k(\pi)+k.$$

For a multivariate Gaussian distribution $\pi$, the tractable form of $\mathbb{E}_{x\sim\pi}\big[-f'(x)+\varphi(x)\big]$ can be derived by using that of Tsallis entropy $\mathcal{T}_q^k(\pi)$ of $\pi$. Thus $\psi_\pi$ can be rewritten as

$$\psi_\pi(s,a) = q\varphi(s)+(q-1)\mathcal{T}_q^k(\pi)$$

In the special case of $q=1$ and $k=1$, we have $\psi_\pi(s,a)=\log\pi(a|s)$.

## B.3  BREGMAN DIVERGENCE WITH TSALLIS ENTROPY REGULARIZATION

We consider the following form of the Bregman divergence:

$$\int \pi(x)\Big\{f'\big(\hat{\pi}(x)\big)-\omega\big(\pi(x)\big)\Big\}\,\mathrm{d}x - \int \hat{\pi}(x)\Big\{f'\big(\hat{\pi}(x)\big)-\omega\big(\hat{\pi}(x)\big)\Big\}\,\mathrm{d}x$$

For $\omega(x)=\frac{k}{q-1}(1-x^{q-1})$, $f'(x)=\frac{k}{q-1}(1-qx^{q-1})=q\omega(x)-k$, and $k=1$, the above form is equal to

$$\int \pi(x)\left[\frac{1-q\hat{\pi}(x)^{q-1}}{q-1}\right]\mathrm{d}x - \mathcal{T}_q(\pi)-(q-1)\mathcal{T}_q(\hat{\pi})+1$$

$$= \frac{1}{q-1}-\frac{q}{q-1}\int \pi(x)\hat{\pi}(x)^{q-1}\,\mathrm{d}x - \mathcal{T}_q(\pi)-(q-1)\mathcal{T}_q(\hat{\pi})+1$$

$$= \frac{q}{q-1}-\frac{q}{q-1}\int \pi(x)\hat{\pi}(x)^{q-1}\,\mathrm{d}x - \mathcal{T}_q(\pi)-(q-1)\mathcal{T}_q(\hat{\pi}).$$

Let us define two multivariate Gaussians as follows:

$$\pi(x)=\mathcal{N}(x;\mu,\Sigma),\ \mu=[\mu_1,\cdots,\mu_d]^\mathsf{T},\ \Sigma=L\,\mathrm{diag}(\sigma_1^2,\cdots,\sigma_d^2)L^\mathsf{T},$$

$$\hat{\pi}(x)=\mathcal{N}(x;\hat{\mu},\hat{\Sigma}),\ \hat{\mu}=[\hat{\mu}_1,\cdots,\hat{\mu}_d]^\mathsf{T},\ \hat{\Sigma}=\hat{L}\,\mathrm{diag}(\hat{\sigma}_1^2,\cdots,\sigma_d^2)\hat{L}^\mathsf{T},$$

where $L$ and $\hat{L}$ denote unit lower triangular matrix. We have

$$\int \pi(x)\hat{\pi}(x)^{q-1}\,\mathrm{d}x = I(\pi,\hat{\pi};1,q-1)=\exp\big\{F(\theta')-F(\theta)-(q-1)F(\hat{\theta})\big\},$$

where

$$\theta = \begin{bmatrix}\Sigma^{-1}\mu \\ -\frac{1}{2}\Sigma^{-1}\end{bmatrix}$$

$$\hat{\theta} = \begin{bmatrix}\hat{\Sigma}^{-1}\mu \\ -\frac{1}{2}\hat{\Sigma}^{-1}\end{bmatrix}$$

$$\theta' = \theta+(q-1)\hat{\theta} = \begin{bmatrix}\Sigma^{-1}\mu+(q-1)\hat{\Sigma}^{-1}\mu \\ -\frac{1}{2}(\Sigma^{-1}+(q-1)\hat{\Sigma}^{-1})\end{bmatrix} = \begin{bmatrix}\theta_1' \\ \theta_2'\end{bmatrix}$$

and

$$F(\theta) = \frac{1}{2}\mu^\mathsf{T}\Sigma^{-1}\mu+\frac{1}{2}\ln(2\pi)^d|\Sigma| = \frac{1}{2}(\mu)^\mathsf{T}\Sigma^{-1}\mu+\sum_{i=1}^d\frac{\ln 2\pi}{2}+\ln\sigma_i,$$

$$F(\hat{\theta}) = \frac{1}{2}\hat{\mu}^\mathsf{T}\hat{\Sigma}^{-1}\hat{\mu}+\frac{1}{2}\ln(2\pi)^d|\hat{\Sigma}| = \frac{1}{2}(\hat{\mu})^\mathsf{T}\hat{\Sigma}^{-1}\hat{\mu}+\sum_{i=1}^d\frac{\ln 2\pi}{2}+\ln\hat{\sigma}_i,$$

$$F\big(\theta+(q-1)\hat{\theta}\big) = -\frac{1}{4}(\theta_1')^\mathsf{T}(\theta_2')^{-1}(\theta_1')+\frac{1}{2}\ln|-\pi(\theta_2')^{-1}|$$

We can replace some difficult computations using LDL decomposition such as $\Sigma^{-1} = L^{-1}\,\mathrm{diag}(1/\sigma)(L^{-1})^\mathsf{T}$ and $|\Sigma|=\sum_{i=1}^d\ln\sigma_i$.

## C IMPLEMENTATION DETAILS

Unnormalized rewards of the IRL algorithm often mislead the agent to take unnecessary awareness of *termination* in finite-horizon MDPs (Kostrikov et al., 2018). To solve the issue IRL algorithms need to remove the difference between regarding steps depending on MDP's time. Doob's optimal stopping theorem formally states that the expected value of a martingale at a stopping time is equal to its initial expectation. Assume a martingale makes the entire procedure as a fair game on average, which means nothing can be gained by stopping the play.

**Theorem 4** (Doob's optimal stopping theorem). *Let a process $\{X_t\}_{t=1}^{\infty}$ be a martingale and $\tau$ be a stopping time with respect to filtration $\{\mathcal{F}_t\}_{t \geq 1}$. Assume that one of the conditions holds:*

*(a) $\tau$ is almost surely bounded, i.e., there exists a constant $c \in \mathbb{N}$ such that $\tau \leq c$.*

*(b) $\tau$ has finite expectation and the conditional expectations of the absolute value of the martingale increments almost surely bounded, more precisely, $\mathbb{E}[\tau] < \infty$ and there exists a constant $c$ such that $\mathbb{E}\big[|X_{t+1} - X_t|\,\big|\mathcal{F}_t\big] \leq c$ almost surely on the event $\{\tau > t\}$ for all $t \geq 0$.*

*(c) There exists a constant $c$ such that $|X_{\min\{t,\tau\}}| \leq c$ almost surely for all $t \geq 0$. Then $X_\tau$ is an almost surely well-defined random variable and $\mathbb{E}[X_\tau] = \mathbb{E}[X_0]$.*

*Then $X_\infty$ is integrable and $\mathbb{E}[X_\infty] = \mathbb{E}[X_0]$*

Doob's optimal stopping theorem states one of the necessary conditions of IRL reward of normalizing the reward measures and making them as a martingale even for finite-horizon benchmarks.

### C.1 NETWORK ARCHITECTURES

For all networks, we use networks with 2-layer MLP with 100 hidden units, respectively. We consider the reward model with two separate neural networks $(\psi_\phi, b(\cdot))$ for the proposed reward function for $\lambda > 0$:

$$r_\phi(s, a) = \psi_\phi^\lambda(s, a) = \lambda\psi_\phi(s, a) + b(s),$$

Motivated by RAIRL-DBM, we consider the reward models in Figure 9. The model outputs reward for proximal updates trained by mirror descent and state-only discriminator network. Discriminating state visitation by $b(\cdot)$ is required because the reward function needs to consider every state (especially the state that cannot be visited by $\pi_E$) until $D_{\mathrm{KL}}(\rho_\pi \| \rho_{\pi_E}) \approx 0$. Figure 9 (a) shows logits of the softmax distribution involved when calculating rewards when the action space is discrete. For continuous control (Figure 9 (b)), the architecture is similar, where the mean and covariance are used to compute a reward for a certain action.

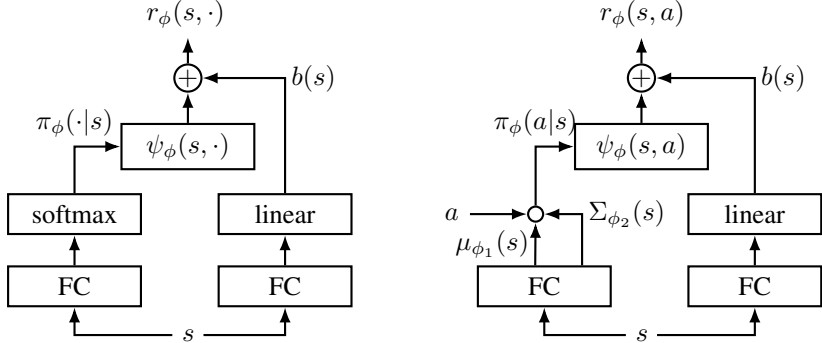

Figure 9: Schematic illustrations of MD-AIRL reward models for discrete (left) and continuous control (right)

### C.2 MULTI-GOAL ENVIRONMENT

Let the 2D coordinate denote the position of a point mass on the environment. In the multi-goal environment, the agent is initially located according to the normal distribution $\mathcal{N}(\mathbf{0}, (0.1)^2\mathrm{I})$. The four goals are located at $(6, 0)$, $(-6, 0)$, $(0, 6)$, and $(0, -6)$, where the agent can move a maximum of

1 unit per timestep for each coordinate. The ground-truth reward is given by the difference between successive values of a Gaussian mixture depicted as Figure 10.

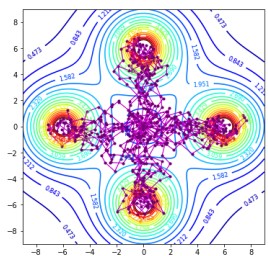

Figure 10: Visualization of the multi-goal environment.

We used the full covariance Gaussian distribution in this experiment (as well as the conceptual experiment in Figure 2. Note that the covariance matrix is positive-definite and symmetric. To achieve numerically stable computation, we applied LDL decomposition to covariance matrix involving unit lower- and upper-triangle matrices, and a diagonal matrix. As a result, the policy network outputs a vector $(\boldsymbol{\mu}(s)^{\mathsf{T}}, \boldsymbol{\sigma}(s)^{\mathsf{T}}, \boldsymbol{l}(s)^{\mathsf{T}})^{\mathsf{T}}$ where the additional part $\boldsymbol{l}(s)$ denotes $\frac{d(d-1)}{2}$ entries of unit lower triangular matrix. Denote $\boldsymbol{L}(s)$ as a unit lower triangular matrix from $\boldsymbol{l}(s)$. For example, the covariance matrix can be reconstructed by

$$\boldsymbol{\Sigma}(s) = \boldsymbol{L}(s)[\mathrm{diag}(\boldsymbol{\sigma}(s))]\boldsymbol{L}(s)^{\mathsf{T}}.$$

Additionally, the action samples can be calculated by

$$a = \boldsymbol{\mu}(s) + \boldsymbol{L}(s)(\boldsymbol{\sigma}(s) \cdot z) \quad z \sim \mathcal{N}(\mathbf{0}, \mathrm{I})$$

Computing inverses, determinants and multiplications with unit triangular matrices and diagonal matrices can be efficiently performed by numerical libraries. Therefore, we can fully model the Bregman divergence and reward using neural networks as provided in Appendix B.

### C.3 MuJoCo Experiments

Instead of directly using squashed policies proposed in SAC (Haarnoja et al., 2018), we assume the application of `tanh` as a part of the environment (known as *hyperbolized* environments of RAIRL (Jeon et al., 2020)). Specifically, after an action $a$ is sampled from the policies, we pass $\mathtt{tanh}(a/1.01) * 1.01$ to the environment. We additionally clip the hyperbolized actions to 1, if the environment is not tolerant to the excessive values of action. Therefore, we can consider the standard diagonal Gaussian policy

$$\pi(\cdot|s) = \mathcal{N}(\boldsymbol{\mu}(s), \boldsymbol{\Sigma}(s))$$

where $\boldsymbol{\mu}(s)$ denotes means $[\mu_1(s), \mu_1(s), \ldots, \mu_d(s)]^{\mathsf{T}}$ and $\boldsymbol{\sigma}(s)$ denotes standard variance $[\sigma_1(s), \sigma_2(s), \ldots, \sigma_d(s)]^{\mathsf{T}}$. We soft-clip the standard deviation as $\sigma_i(s) \in [\ln 0.01, \ln 2]$ for the stability using `tanh`. Last, we update the moving mean of intermediate values of regularized reward $r(s, a) + \lambda\varphi(\pi(a|s))$ and update the RL algorithm with mean-subtracted rewards. In addition to the analyses of (Fu et al., 2017; Jeon et al., 2020) regarding reward shaping and normalization, mean-zero rewards for training agents have the additional property of preventing termination awareness, as stated by the optimal stopping theorem (Theorem 4).

## C.4 HYPERPARAMETERS

Tables 2, 3, and 4 show the hyperparameters of the conducted experiments.

Table 2: Hyperparameters for bandit environments.

| Parameter | Value |
|---|---|
| Learning rate (policy) | $1 \cdot 10^{-3}$ |
| Learning rate (reward) | $1 \cdot 10^{-3}$ |
| $\alpha_1$ | 0.5 |
| $\alpha_T$ | 5 |
| $\lambda$ | 1 |
| Discount factor ($\gamma$) | 0.0 |
| Batch size | 16 |
| Steps per update | 50 |
| Total steps | 300,000 |

Table 3: Hyperparameters for Mmlti-goal environment.

| Parameter | Value |
|---|---|
| Learning rate (policy) | $5 \cdot 10^{-4}$ |
| Learning rate (reward) | $5 \cdot 10^{-4}$ |
| Replay size | 10,000 |
| $\alpha_1$ | 1 |
| $\alpha_T$ | 10 |
| $\lambda$ | 1 |
| Discount factor ($\gamma$) | 0.5 |
| Batch size | 512 |
| Steps per update | 50 |
| Total steps | 300,000 |

Table 4: Hyperparameters for MuJoCo environments.

| Parameter | Value |
|---|---|
| Learning rate (policy) | $3 \cdot 10^{-4}$ |
| Learning rate (reward) | $3 \cdot 10^{-4}$ |
| Replay size | 1,000,000 |
| $\alpha_1$ | 1.0 |
| $\alpha_T$ | 20.0 |
| $\lambda$ | 0.01 |
| Discount factor ($\gamma$) | 0.99 |
| Batch size | 256 |
| Steps per update | 5 |
| Initial exploration | 10,000 |
| Total steps | 1,000,000 |

