# OpenReview forum: "Robust Imitation via Mirror Descent Inverse Reinforcement Learning"
_ICLR.cc/2022/Conference — ICLR 2022 Submitted_

### Official Review · Reviewer_c6kk · 2021-10-30

**Correctness:** 3
**Technical Novelty And Significance:** 3
**Empirical Novelty And Significance:** 3
**Recommendation:** 5
**Confidence:** 4

**Main Review:**

Strengths:

1. The idea of using MD for reward updates is novel. A detailed theoretical guarantee on sufficiency and necessities on the choices of stepsizes are provided.

2. The paper tends to develop MD-AIRL step by step, and also provides illustrative figures to help to understand.

Weaknesses:

My complaints most come from the writing and illustration part. Please see the detailed comments as follows.

1. Figure 1: what does the solid curve in policy/reward spaces mean? What about the dotted angles? The authors are encouraged to elaborate more on the illustration.

2. “For instance, $\psi_{t+1}$ can represent a softmax policy for a discrete space, ...”: is $\psi_{t+1}$ a regularized reward function or a policy? I am assuming that it should be a reward function.

3. Equation 5: It is a maximization problem of $\pi$ but I cannot find $\pi$ in the objective. Also, I am not clear how the form of $J_\Omega(\pi_t, \psi_{t+1})$ is derived in equation 5, and how it is related to equation 2. The authors are encouraged to elaborate more to make it clear.

4. Below equation 8: Is $\pi_*$ a function of $t$? What do the authors mean by “the expectation is taken over the entire steps”? How is the optimality condition $E[...] = 0$ is derived? I am confused about the whole paragraph.

5. Theorem 1: When will the assumptions on $\Omega$ in theorem 1 be satisfied?  Are there any examples? I am thinking about KL but it does not satisfy the conditions here. Please correct me if I am wrong.

6. Theorem 2: The last inequalities involve $s$. What is this $s$ here?

7. Proof of Theorem 2: what is $n$ below equation 33? I am not sure about the relation between $f_t$ and $D_\Omega$ throughout the proof.

8. Proof of Lemma 6: The authors state "Assuming all states are reachable". Is this an assumption of the results? What if such a statement does not hold?


Minor issues:

1. Section 8: functinos -> functions

2. Above Lemma 1: greed manner -> greedy manner

3. Below Figure 1: a update -> an update

4. Equation 7: gradient of $D_\Omega$ is a bit misleading; the authors should mention that the gradient is taken w.r.t. the first argument.

5. Equation 8: $\Omega(\pi(\cdot | s_i))$ -> $\Omega(\pi_t(\cdot | s_i))$


**Summary Of The Paper:**

The paper proposes an algorithm MD-AIRL, where each iteration involves policy and reward updates, and an expert policy estimation. The algorithm is motivated by MD, especially the reward update is modeled as a projected update of an MD update. The sufficiency and necessities of the choices of stepsizes to guarantee convergence are theoretically justified. A practical algorithm with numerical experiments are also provided.

**Summary Of The Review:**

Overall I think the paper proposes a novel approach to IRL. But the most obvious drawback is that the paper is not clearly written, which has many significant issues leading readers to misunderstand/cannot understand the paper. I think the current revision is not sufficient to be published in ICLR, thus I recommend a marginal reject and tend to reject if there is no significant revision.

---

> ### Author Response · Authors · 2021-11-15
> **Response to Reviewer 3**
>
> We thank **R3**'s fruitful questions and comments.
>
> ## Response to Comments and Questions
>
> > Figure 1: The authors are encouraged to elaborate more on the illustration.
>
> Thank you for your suggestion. We would like to elaborate that the two solid lines in the left (policy space) indicate the constraint of probability simplex and the proximity term, respectively. The dotted angle (or curve) denotes a Bregman projection phase which is omitted thanks to Lemma 1. In the revised version, we fixed the illustration and rewrote the caption entirely.
>
> > "For instance, $\psi_{t+1}$ can represent a softmax policy for a discrete space, ...": is $\psi_{t+1}$ a regularized reward function or a policy? I am assuming that it should be a reward function.
>
> We agree that this statement would confuse readers. To summarize, $\psi_{t+1}$ is a reward function, but it can be transformed into a policy by using its parameters. We fixed this issue in the revised version.
>
> > Equation 5: The authors are encouraged to elaborate more to make it clear.
>
> Thank you for this point; we noticed that the maximizing argument $\pi$ was missing in the objective function. We substituted $J_\Omega(\pi_t, \psi_{t+1})$ to $J_\Omega(\pi, \psi_{t+1})$ and enhanced readability of related lines.
>
> > Theorem 1: When will the assumptions on $\Omega$ in theorem 1 be satisfied? Are there any examples? I am thinking about KL but it does not satisfy the conditions here. Please correct me if I am wrong.
>
> Yes, indeed, the KL divergence is one of the outliers of this argument when we start to consider extreme cases of distributions (i.e., $p_i \approx 0$ for certain elements of a distribution $p$). Therefore, in practice, we have to enforce bound for the feasible action space and restrict parameters to make every element have non-zero probabilities. A more fundamental theoretical analysis of these outliers remains as future work, and we added this argument in Section 8.
>
> > Theorem 2: What is this $s$ here?
>
> There was an error in the last statement of Theorem 2. We corrected this line to "If $\eta_t \equiv \eta_1$, then there exist $c_1,c_2\in(0,1)$ such that $c_1 \cdot A_1 \le A_T \le c_2 \cdot A_1$, where $(A_t)$ denote a sequence of $A_t=\mathbb{E}_{\tau_1 \dots \tau_t}\bigl[D_\Omega(\pi_\ast\Vert\pi_t)\bigr]$". This issue is now resolved in the revised manuscript.
>
> > Proof of Theorem 2: what is $n$ below Equation 33? I am not sure about the relation between $f_t$ and $D_\Omega$ throughout the proof.
>
> We agree that this line has a clarity issue. The number $n\in\mathbb{N}$ is an arbitrary, sufficiently large number that makes the scalar term $(1 - 2L\omega^{-1}\eta_t)$ to be less than 1 for $t>n$, since $\eta_t$ is guaranteed to be converged to 0 as $t\to\infty$ by the previous arguments. We fixed this line in the revised manuscript. $f_t$ is an online cost function that approximates the global cost function using a sampled trajectory $\tau_t$ and a reward function. By considering, the convergence of $f_t$, the convergence of a discounted sum of Bregman divergences are naturally driven.
>
> > Proof of Lemma 6: Is this an assumption of the results? What if such a statement does not hold?
>
> The intention of the statement "assuming all states are reachable" is simplified description of an assumption of the decomposability condition (Fu et al. 2018, Definition B.1). To the best of our knowledge, this condition is a common premise in the AIRL domain. In realistic problems, there can be unreachable states depending on initial states and the given time limits. However, we believe that this condition does not substantially deteriorate the generality of our theoretical arguments.
>
> ## Response to Minor Issues
>
> Thank you for letting us know typos and unclear derivations in the original manuscript. This information helped our work on improving its clarity. Please check our **Official Revision** where we fixed these issues.
>
> ## Reference
>
> * Justin Fu, Katie Luo, and Sergey Levine. Learning robust rewards with adversarial inverse reinforcement learning. ICLR 2018.

---

### Official Review · Reviewer_fhXG · 2021-11-01

**Correctness:** 3
**Technical Novelty And Significance:** 3
**Empirical Novelty And Significance:** 2
**Recommendation:** 5
**Confidence:** 3

**Main Review:**

Pros:
1. The proposed MD-AIRL are both theoretically and empirically justified. Theoretically, the reward derived by mirror decent algorithm ensures the minimization of Bregman divergence converges to local optima along with a rigorous regret bound. Empirically, the resulting MD-AIRL was evaluated with large discrete action spaces and continuous action spaces.
2. It is good to see the dense theoretical analysis in the paper for introducing the MD-AIRL gradually. The MD can be implemented on top of existing adversarial imitation learning method.

Cons:
1. This paper claims that the proposed Mirror Descent Inverse Reinforcement Learning is more robust than previous adversarial imitation learning method. However, in the experiments, there lacks evidence to support that claim.
2. In Figure 8, the results show that the proposed MD-AIRL using Tsallis regularizer with p=2 is competitive or slightly better than RAIRL. How does the MD-AIRL perform with other choices of p and regularizer?
3. RAIRL and MD-AIRL both use Tsallis regularizer, however, the relationship between the two methods has not been well discussed.



**Summary Of The Paper:**

This paper presents a novel mirror-descent adversarial inverse reinforcement learning (MD-AIRL) algorithm. MD-AIRL considers the reward function as an iterative sequence in a proximal method. MD-AIRL has been introduced with dense theoretical analysis and validated with diverse experiments covering both discrete and continuous action spaces.

**Summary Of The Review:**

Despite its novelty and dense theoretical analysis, the empirical evaluation can be further improved to better support the robustness of MD-AIRL. I also hope to see more discussion between MD-AIRL and the related RAIRL in the rebuttal. I read through both authors' responses and other reviews and decided to recommend a marginal reject. I have thought the submission is interesting and has its own contribution in some extent, but it seems like there haven't been signficant improvement during the rebuttal period. The performance improvement is not significant and the written clarity is not ready to be acceoted by ICLR.

---

> ### Author Response · Authors · 2021-11-15
> **Response to Reviewer 2**
>
> We appreciate **R2** for the thoughtful feedback. We are encouraged by overall positive comments on our theoretical analyses. Here we provide responses to your questions.
>
> > RAIRL and MD-AIRL both use Tsallis regularizer, however, the relationship between the two methods has not been well discussed.
>
> We agree that the initial draft needs improvements in writing in terms of highlighting the difference between the two methods. Both algorithms can use Tsallis entropy as a choice of regularizers, but MD-AIRL additionally uses MD formulation (Equation 14) in its learning. Both algorithms are highly general in terms of choice of regularizer, but our method additionally provides good theoretical properties originated from optimization research.
>
> > The empirical evaluation can be further improved to better support the robustness of MD-AIRL.
>
> We agree that there is likely a better choice of the step size for each task if further hyperparameter searches have been conducted. as the choice of hyperparameters is motivated by the initial toy experiment described in Figure 2 from Section 3. However, we would like to point out that our contributions are not mainly on significantly outperforming a specific task, and yet we respectively emphasize that MD-AIRL achieves consistent gains throughout the various settings.
>
> > Figure 8: How does the MD-AIRL perform with other choices of $q$ and regularizer?
>
> The overall choices of experimental settings (number of demonstrations and values fo $q$) are summarized in Figure 7.

---

### Official Review · Reviewer_4kT4 · 2021-11-08

**Correctness:** 4
**Technical Novelty And Significance:** 2
**Empirical Novelty And Significance:** 1
**Recommendation:** 5
**Confidence:** 3

**Main Review:**

This paper seems to have an incremental contribution to AIRL and it claims AIRL benefits from Mirror Descent updates. It is incremental in a sense that  MD has been studied extensively in optimization research. However, providing some intuitions on the benefits of MD in IL might be interesting for some researchers.

Strength:
- Provided convergence and regret bound
- Better performance on experiments (slightly)

The derivations seem correct, but I did not check them thoroughly.

Weakness:
- The presentation of the paper needs improvement, especially on the connection of the previous works and the proposed method.
- It is unclear how MD affects the resulted reward function (or the corresponding policy) for examples as opposed to SGD?
- Experiments are not sufficient as authors compared the proposed method only with one other imitation learning method (RAIRL).

Detailed Review:


From the presentation it it not clear for me the significance of the contribution. More precisely:
- It seems the step size greatly affects the convergence rate of the proposed method. So does this method require properly tuning the step-size for each problem? Is there experiments on the effect of step-size on the produced policy/reward function?
How the choice of SGD vs MD affects the reward function?
- "A gradient may not be the direction of the steepest descent in this case due to geometric constraints" How to verify this claim in IL setting, faster convergence or superior reward function?  At convergence and without any prior knowledge on the optimal step size, can you claim that it is expected that MD will result in a higher quality reward function as opposed to other AIRL methods?
-Can you discuss more on how and why the choice of MD will resolve the two mentioned issued in AIRL? Can it be shown in the experiments?
-Can you explain in details the claim "We verified that the proposed method has clear advantages over previous AIL methods in terms of robustness"? Is it verified through multi-armed bandit experiment?




**Summary Of The Paper:**

This paper proposes an alternative optimization technique (Mirror Descent) for adversarial inverse reinforcement learning, claiming it will resolve some of the issues with previous methods.

-The authors prove the convergence of the proposed optimization method and provide a regret bound.
-They experimentally show the benefits of the proposed methods in discrete and continuous action spaces.



**Summary Of The Review:**

Overall I think this is an interesting paper. However, my main concern is the accuracy of the claims and the presentation of the paper.

---

> ### Author Response · Authors · 2021-11-15
> **Response to Reviewer 1**
>
> We thank **R1** for constructive feedback.
>
> > MD has been studied extensively in optimization research.
>
> It is true that MD has been one of the important topics and extensively studied in the optimization research. Nevertheless, we respectively emphasize that our contributions include, but are not limited to, reintroducing MD. We provided a new interpretation between policies and regularized reward functions. We also newly introduced our online imitation learning setting, which allows a much more general condition on the adversarial learning (including non-convergent behavior of discriminators) than previous methods. We believe our work contributes to AIL researches that lacks a connection to theoretical foundations originated from optimization studies.
>
> > "A gradient may not be the direction of the steepest descent" How to verify this claim in IL setting, faster convergence, or superior reward function?
>
> We would like to elaborate on the underlying postulation that a policy, a collection of conditional probability distributions, is not in the Euclidean space but in a statistical manifold. This implies that we have to consider geodesics when measuring distances and certain gradient flows in order to calculate the steepest descent. MD is a discretization of computing a Riemannian gradient flow (Gunasekar et al. 2020) with a finite step size where the Bregman divergence plays a role of an approximated distance-like function.
>
> > It is unclear how MD affects the resulted reward function (or the corresponding policy) for examples as opposed to SGD?
>
> * Mirror descent is a _template_ first-order optimization method for optimizing a cost function with respect to a geometry specified by a convex potential function $\Omega$ (Nemirovsky & Yudin, 1983). Accordingly, various existing optimization methods can be considered as MD, and SGD is indeed a representative MD algorithm where the geometry is specified by the Euclidean geometry (i.e., $\Omega(x) = \tfrac{1}{2}\lVert x\rVert^2_2$).
> * Specifically, our work mainly considers iterative policies that are in a space of probability distributions ($\Delta^\mathcal{S}_\mathcal{A}$ in Figure 1). For distributions, MD with the negative Tsallis entropy with $q=2$ ($\Omega(p) = -\mathcal{T}^1_2(p) = \tfrac{1}{2}\sum_i p^2_i - \tfrac{1}{2}$) can be considrered as (conceptually) the most similar method to SGD.
> * We proposed a method that incorporates various regularizers. We think the optimal $\Omega$ really depends on each environment settings and how the stochastic estimation process of expert distribution ($\bar{\pi}_{E,t}$) forms, since we observed that the choice of $\Omega$ affects performance differently depending on experiments. Typically, identifying an underlying geometry for a specific problem requires prior knowledge. Finding an efficient way of knowing this information only with a few trials apparently remains as a future work.
>
> > Experiments are not sufficient as authors compared the proposed method only with one other imitation learning method (RAIRL).
>
> We would like to clarify that our motivation and contribution are not primarily in outperforming the baselines with specific regularization functions. Our method is based on the generalized perspective on IRL, which is a similar generalization level to RAIRL in terms of the choice of $\Omega$. To the best of our knowledge, there is not much choice of highly-general and performant IRL algorithms to conduct comparative studies. We reported that MDIRL brings consistent gains over the RAIRL approach throughout various experiments.
>
> > Can you discuss more on how and why the choice of MD will resolve the two mentioned issued in AIRL?
>
> 1. For the first issue, we adopted MD, an optimization framework with good theoretical properties. We demonstrated that our work is applicable to various tasks with five different convex regularizers.
> 2. The second (robustness) is related to experiments where the demonstration data are not sufficient to learn a fixed form of divergence. In this work, we consider the robustness of the imitation learning algorithm to be closely related to the situation of insufficient data. This is verified through the multi-armed bandit experiment with large-scale action space and the MuJoCo experiment with small demonstrations (Table 1 and Figure 7).
>
> > The presentation of the paper needs improvement.
>
> We agree that the original manuscript lacks clarity in some parts. Please check **Official Revision** and let us know whether your concerns are relieved.
>
> ## References
>
> * Nemirovskij, Arkadij Semenovič, and David Borisovich Yudin, Problem Complexity and Method Efficiency in Optimization. A Wiley-Interscience Publication. Wiley, 1983.
> * Suriya Gunasekar, Blake Woodworth, and Nathan Srebro. Mirrorless mirror descent: A More Natural Discretization of Riemannian Gradient Flow. arXiv preprint arXiv:2004.01025, 2020.

---

> > ### Author Response · Authors · 2021-11-15
> > **Extended Experimental Results**
> >
> > > Is there experiments on the effect of step size on the produced policy/reward function?
> >
> > Yes, here we show extended results of the toy problem in Figure 2.
> >
> > **Table.** Bregman divergences after the final MD steps for different step size scheduling ($T=100$).
> >
> > |($\eta_1$, $\eta_T$)| Shannon | Tsallis, $q=1.1$ | Tsallis, $q=1.5$ | Tsallis, $q=2.0$ |
> > |-|---------|---------|---------|---------|
> > |(2,2)|-|1.20±0.64|0.34±0.26|0.69±0.46|
> > |(1,1)|0.086±0.080|0.15±0.25|0.22±0.32|0.11±0.17|
> > |(0.5,0.5)|0.067±0.060|0.076±0.069|0.038±0.046|0.060±0.058|
> > |(0.2,0.2)|0.0105±0.0091|0.032±0.032|0.024±0.041|**0.018±0.016**|
> > |**(1,0.1)**|**0.0071±0.0086**|**0.0093±0.011**|**0.017±0.011**|0.021±0.009|
> > |(1,0.01)|0.015±0.015|0.011±0.014|0.021±0.021|0.033±0.019|
> >
> > The table shows that scheduling $\eta_t$ with a harmonic progression $\eta_1=1$ $\eta_T=0.1$ shows overall good results in this experiment. From these results and our theoretical arguments, MD is recommended to gradually lower $\eta_t$ (so $\lim_{t\to\infty}\eta_t = 0$) as described in Equation (10). Also, there is optimal scheduling of the step size $\eta_t$, depending on $\Omega$ and $T$. As a rule of thumb, we recommend setting the initial step size as $\eta_1\le 1$ and scheduling to $\eta_T \approx 0$ if $T$ is sufficiently large for training.

---

### Author Response · Authors · 2021-11-15
**Response to All Reviewers**

We thank the three reviewers for their efforts and insightful reviews. We are encouraged that they find our motivation and idea to be novel (**R2**, **R3**) and interesting (**R1**). We are glad all reviewers gave positive feedback on our theoretical analyses. Here we curate key responses to important issues and questions.
* **Relationship between MD-AIRL and RAIRL (Jeon et al., 2020).**
We summarize the connection between MD-AIRL and RAIRL as follows.
  * Our work is closely related to the work done by Jeon et al. (2020); Jeon et al. introduced a tractable computation method of the regularized reward operator (Definition 1) for various distributions with respect to a strongly convex function $\Omega$, which made the beginning of our arguments possible. However, we introduced a new proximal optimization method in the adversarial imitation learning (AIL) research, motivated by mirror descent (MD) and Bregman divergences' basic geometric properties.
  * Considering each algorithm as an optimization process that outputs a trained reward function after the finite step $T$, both methods produce reward functions with similar neural network configurations and work for a given convex function $\Omega$.
  * Although MD-AIRL incorporates adversarial learning in its training process, MD-AIRL additionally applies an MD formulation (Equation 14) and outputs a more stable form of reward functions in terms of regrets. One noteworthy advantage of our approach is that it allows a much more general condition on the adversarial learning (including non-convergent behavior of discriminators) than previous methods.
* **Significance of empirical evidence (R1, R2).**
We respectively emphasize that the primary intention of our experiments was to show that applying MD update rules is complementary to standard imitation learning methods. Thanks to the extensive optimization studies, we provide an original interpretation of the RL-IRL framework that draws good theoretical properties. We believe that our work mitigates instability issues often reported from adversarial learning researches.
As each experiment controls all irrelevant hyperparameters, MD-AIRL achieves consistent gains over the RAIRL approach. Our empirical evidence can add to advancement in the AIL research field; the results match our theoretical analyses, and the experiments cover various settings and environments (discrete and continuous). We believe results from MD-AIRL can bring meaningful progress to researches of imitation learning for sequential decisions.
* **Clarity of paper (R1, R3).**
We agree that that the readability of the paper needed enhancement. In addition to **Official Response** for each reviewer, we posted **Official Revision** where we fixed weaknesses in the original submission pointed out by the reviewers. In the revision, we left margin notes (`FIX`/`NEW` keyword along with the reviewers' IDs) indicating modified parts. Please let us know if there are further questions during the rest of the discussion phase.

---

### Decision · Program_Chairs · 2022-01-20

**Decision:**

Reject

**Comment:**

This work introduces/applies the mirror descent optimization technique to adversarial inverse reinforcement learning (AIRL). As a result, the proposed algorithm (MD-AIRL) incrementally learns a parameterized reward function in an associated reward space. The two issues of standard adversarial imitation learning algorithms are 1) current "divergence"-based updates may not lead to updates that better match the expert (due to geometry) 2) "divergence"-based updates may suffer when only small number of demonstrations are provided. Thus the goal of this work is to (presumably)  to "robustify" the learning of reward function especially by addressing these issues. The proposed algorithm is evaluated on a bandits problem, a multi-goal toy example and standard mujoco benchmark.

**Strengths**
This work attempts to address the important problem of understanding and improving the updates of IRL algorithms
A theoretical analysis is provided

**Weaknesses**
The major concern is clarity of the manuscript. Even after updating clarity remains a concern
While a lot of experiments were performed, evaluation is not entirely convincing. One reason for this that it is hard to tie the results back to the original motivation/claims of this algorithm. As one reviewer notes "it's unclear how the new algorithm affects reward functions". Furthermore, reviewers find the experimental results not entirely convincing

**Summary**
After rebuttal and revision, the clarity and experimental analysis remain a concern. My recommendation is that the authors are encouraged to take the reviewers feedback and improve the manuscript. In its current form it's not quite ready yet for publication.